# The ATPases of cohesin interface with regulators to modulate cohesin-mediated DNA tethering

**Gamze Çamdere, Vincent Guacci\*, Jeremiah Stricklin, Douglas Koshland\***

Department of Molecular and Cell Biology, University of California, Berkeley, Berkeley, United States

**Abstract** Cohesin tethers together regions of DNA, thereby mediating higher order chromatin organization that is critical for sister chromatid cohesion, DNA repair and transcriptional regulation. Cohesin contains a heterodimeric ATP-binding Cassette (ABC) ATPase comprised of Smc1 and Smc3 ATPase active sites. These ATPases are required for cohesin to bind DNA. Cohesin's DNA binding activity is also promoted by the Eco1 acetyltransferase and inhibited by Wpl1. Recently we showed that after cohesin stably binds DNA, a second step is required for DNA tethering. This second step is also controlled by Eco1 acetylation. Here, we use genetic and biochemical analyses to show that this second DNA tethering step is regulated by cohesin ATPase. Furthermore, our results also suggest that Eco1 promotes cohesion by modulating the ATPase cycle of DNA-bound cohesin in a state that is permissive for DNA tethering and refractory to Wpl1 inhibition.

## Introduction

Multi-subunit protein complexes called Structural Maintenance of Chromosomes (SMC) mediate most aspects of higher-order chromosome organization (*Hirano, 2006*). Cohesin is an SMC complex that was first identified as being essential for sister chromatid cohesion, the process of holding together the sister chromatids from their synthesis in S phase until their segregation in anaphase (*Guacci et al., 1997*; *Michaelis et al., 1997*). Cohesin is also involved in mitotic condensation, meiotic chromosome condensation and structure, post-replicative DNA repair, and transcriptional regulation (*Onn et al., 2008*). Cohesin performs these different functions by tethering together two regions of DNA, either within a single DNA molecule (condensation and regulation of gene expression) or between two DNA molecules (sister chromatid cohesion and DNA repair). How cohesin-mediated DNA tethering is regulated is one of the critical unanswered questions in the field.

The process of sister chromatid cohesion involves the binding of cohesin to chromosomes prior to DNA synthesis, and the subsequent tethering of sister chromatids by the DNA-bound cohesin to generate cohesion during S phase (*Onn et al., 2008*). Cohesin appears to topologically entrap DNA through its ring-like architecture that results from the assembly of its four core subunits, Smc1, Smc3, Mcd1/Scc1, and Scc3/Irr1 (*Nasmyth and Haering, 2009*) (*Figure 1A*). Cohesin's loading onto DNA is controlled by its ATP Binding Cassette (ABC) ATPase domain (*Arumugam et al., 2003*; *Heidinger-Pauli et al., 2010b*; *Murayama and Uhlmann, 2014*). Cohesin's ATPase domain is composed of two ATPase active sites each containing four conserved motifs: Walker A, Walker B, signature motif, and D-loop. The Smc1 ATPase active site is a hybrid site containing the Walker A and Walker B motif s encoded by Smc1, and the D-loop and signature motifs encoded by Smc3. Likewise, the Smc3 ATPase active site contains the Smc3-encoded Walker A and Walker B motifs, and Smc1-encoded D-loop and signature motifs (*Arumugam et al., 2006*; *Haering et al., 2004*; *Hopfner et al., 2000*) (*Figure 1A* inset). Characterization of mutations that block ATP binding

**\*For correspondence:** Guacci@ berkeley.edu (VG); koshland@ berkeley.edu (DK)

**Competing interests:** The authors declare that no competing interests exist.

**eLife digest** The bulk of the genetic material in cells from yeast to humans is organized into chromosomes. These chromosomes must be duplicated and the copies need to be segregated every time cells divide. Cohesin is a protein complex that helps to organize the structure of chromosomes by tethering together two regions of DNA, either within a chromosome or between chromosomes. Problems with cohesin have been linked to cancer and birth defects, but it is not clear how cohesin binds DNA and how it makes a tether between two DNA regions. It is also unclear how cohesin's activity is coordinated with the series of events that allow cells to divide (known as the cell cycle).

Cohesin has two active sites that can break down molecules of ATP. Previous research had suggested that these active sites (called ATPases) controlled cohesin's activity by regulating whether or not it could bind to DNA. However, Çamdere et al. now reveal that cohesin's ATPases do not simply provide an 'on/off switch' for DNA binding. The experiments, which involved a combination of genetic, cell biology and biochemical techniques in budding yeast, instead revealed that one of cohesin's ATPases regulates structural rearrangements in cohesin that is already bound to DNA. These structural rearrangements fine-tune the complex's ability to tether two regions of DNA.

Further experiments then revealed that two cohesin regulators (namely Eco1 and Wpl1) altered this ATPase active site to control cohesin's DNA tethering and DNA binding activities. These findings provide a molecular explanation for how these regulators control cohesin's activity to make sure that the chromosomes have the correct structure during cell division.

The next challenge is to identify the structural changes in cohesin that are triggered by cohesin's two ATPases and to understand how these structural changes promote DNA binding followed by DNA tethering.

(Walker A) or hydrolysis (Walker B) suggest that ATP binding and hydrolysis by both ATPases are necessary for the DNA binding of cohesin (*Arumugam et al., 2003*; *Heidinger-Pauli et al., 2010b*). Cohesin's ATPase activity is also stimulated by the Scc2/Scc4 complex (*Murayama and Uhlmann, 2014*), which is required for the DNA binding of cohesin in vivo (*Ciosk et al., 2000*). These observations led to the hypothesis that Scc2/Scc4 complex stimulates ATP hydrolysis to open the cohesin ring and allow the entry of DNA. Subsequent ATP binding closes the ring, entrapping the DNA (*Arumugam et al., 2003*). Presumably, in order for cohesin to remain stably bound to DNA, its ATPase activity would have to be suppressed to prevent ring reopening and DNA escape. However, the ATPase activity of cohesin in its stable DNA-bound form had never been determined.

Cohesin binding to DNA is also regulated by the antagonistic activites of Wpl1 and Eco1. Wpl1 is thought to destabilize cohesin binding to DNA (*Chan et al., 2012*; *Kueng et al., 2006*). Wpl1 is antagonized by the Eco1 acetyl-transferase through its acetylation of lysines 112 and 113 of Smc3. The key evidence supporting this view is that the deletion of *WPL1 (wpl1Δ)* suppresses the inviability of cells lacking *ECO1 (eco1Δ)* or expressing acetyl null alleles of Smc3 (*smc3-K112R, K113R*) (*Rolef Ben-Shahar et al., 2008*; *Sutani et al., 2009*; *Unal et al., 2008*; *Zhang et al., 2008*).

Recent evidence suggests that Eco1 acetylation promotes cohesion by additional mechanisms besides stabilizing cohesin binding to DNA. First, while *eco1Δ wpl1Δ* cells are viable and have stable cohesin-DNA interaction, they have cohesion defects as severe as mutants of *ECO1* or cohesin subunits (*Guacci and Koshland, 2012*; *Rowland et al., 2009*; *Sutani et al., 2009*). Second, other mutations identified in cohesin and its regulators demonstrate that stable binding of cohesin to DNA is not sufficient for cohesion (*Eng et al., 2014*; *Guacci et al., 2015*). Together, these data strongly argue that cohesion is a two-step process: First, cohesin associates with DNA in a stable form. Then, cohesin undergoes a second transition to tether sister chromatids together. This transition could entail conformational changes involving oligomerization (*Eng et al., 2015*), or the activation of a second, independent DNA binding activity through rearrangements of the coiled coils (*Soh et al., 2015*).

How is cohesin-mediated DNA tethering regulated? One hypothesis is that Eco1-mediated acetylation of Smc3 regulates this second, post-DNA binding step by modulating the cohesin ATPase (*Guacci et al., 2015*). This hypothesis appears to contradict the finding that Walker A and Walker B

mutations in either cohesin ATPase blocks DNA binding (*Arumugam et al., 2003*; *Heidinger-Pauli et al., 2010b*). However, this observation does not preclude a specialized role for the Smc3 ATPase active site in regulating DNA tethering after DNA binding. Indeed, the acetylated K112 and K113 residues in Smc3 are proximal to the Smc3 ATPase active site (*Gligoris et al., 2014*; *Haering et al., 2004*). Moreover, a recently identified suppressor mutation located near the Smc3 ATPase active site bypasses the requirement for Smc3 acetylation in cohesion establishment (*Guacci et al., 2015*). Led by these observations, we reconsider the role of the ATPase domain of cohesin as a potential regulator of the second, post-DNA binding step of cohesion establishment.

Here, we present in vitro and in vivo evidence that the ATPase domain of cohesin plays a role after the initial stable DNA binding of cohesin. We provide evidence suggesting that the Smc1 ATPase active site is involved only in regulating DNA binding, whereas the Smc3 ATPase active site functions in DNA tethering as well as DNA binding. We characterize an Smc3 ATPase active site mutant in *Saccharomyces cerevisiae* that bypasses the requirement for Eco1 acetylation in cohesion generation, and uncouples the level of ATPase activity from cohesin's DNA binding and tethering activities. We propose that cohesin's ATPase has two distinct functions in regulating DNA binding and subsequent DNA tethering. We suggest that Eco1 promotes cohesion by slowing or trapping the ATPase cycle of DNA-bound cohesin to promote a conformation that is permissive for DNA tethering and refractory to Wpl1 inhibition.

## Results

### Cohesin that is stably bound to DNA retains its ATPase activity

Earlier models suggest that cohesin's ATPase head domain is only involved in the initial DNA binding step, and that ATP hydrolysis releases the DNA from cohesin. These models predict that stably DNA-bound cohesin should not show ATPase activity. However, recent literature suggests that Eco1 might promote cohesion by regulating the cohesin ATPase after the stable DNA binding of cohesin. If ATPase activity is required to regulate this second step of cohesion establishment, we should be able to observe ATPase activity for purified cohesin-DNA complexes. To test this possibility, we purified *Schizosaccharomyces pombe* cohesin and loader complexes as described previously (*Murayama and Uhlmann, 2014*). Purified cohesin showed basal ATPase activity that was stimulated by the presence of the loader complex and DNA, and abolished by a Walker A mutation in Smc3 (*Figure 1—figure supplement 1*), similar to published results (*Murayama and Uhlmann, 2014*). Cohesin binding to chromosomes in vivo is enriched at centromeres and at cohesion-associated regions (CARs) along the chromosome arms (*Laloraya et al., 2000*; *Onn et al., 2008*), and is highly salt resistant (*Ciosk et al., 2000*). To purify stable cohesin-DNA complexes that are physiologically relevant, we assembled cohesin on DNA molecules that contained *CARC1* sequence and were coupled by both ends to dynabeads. Cohesin and its loader were incubated with DNA-beads under low salt conditions (25 mM KCl, 25 mM NaCl). The cohesin-DNA bead mix was washed with high salt (500 mM KCl) to remove any free cohesin or cohesin not stably bound to DNA (*Figure 1B*). The cohesin that remained bound to the DNA-beads was then eluted and quantified by Coomassie staining or Western blots. In the presence of the loader, 20% of the input cohesin was bound to DNA-beads after the high salt wash (*Figure 1C,D*). In the absence of the loader, 2-fold less cohesin bound to DNA (*Figure 1D*). Cohesin did not bind to beads that lack DNA (*Figure 1C*). In addition, this stable population of cohesin on DNA-beads could be eluted from the beads by either a restriction enzyme digest or a DNase treatment (*Figure 1—figure supplement 2*). These results suggest that cohesin bound specifically to the DNA that was coupled to beads, and did so in a salt-resistant and loader-inducible manner. These DNA binding features recapitulated the properties of stable cohesin-DNA complexes assembled in vivo (*Ciosk et al., 2000*).

To address whether stably DNA-bound cohesin was an active ATPase, we next assessed whether cohesin in our purified, stable cohesin-DNA complexes (CD-B, *Figure 1B*) retained its ATPase activity. DNA-beads were incubated with cohesin and loader, then washed with high salt buffer to remove free and unstably bound cohesin, as described above. Cohesin-DNA beads (CD-B) were then resuspended in buffer containing ATP to measure ATPase activity of CD-B, compared to basal and loader/DNA-stimulated activities of cohesin. No ATPase activity was detected when cohesin was omitted from the reaction (*Figure 1E* and *Figure 1—figure supplement 1*). As shown before

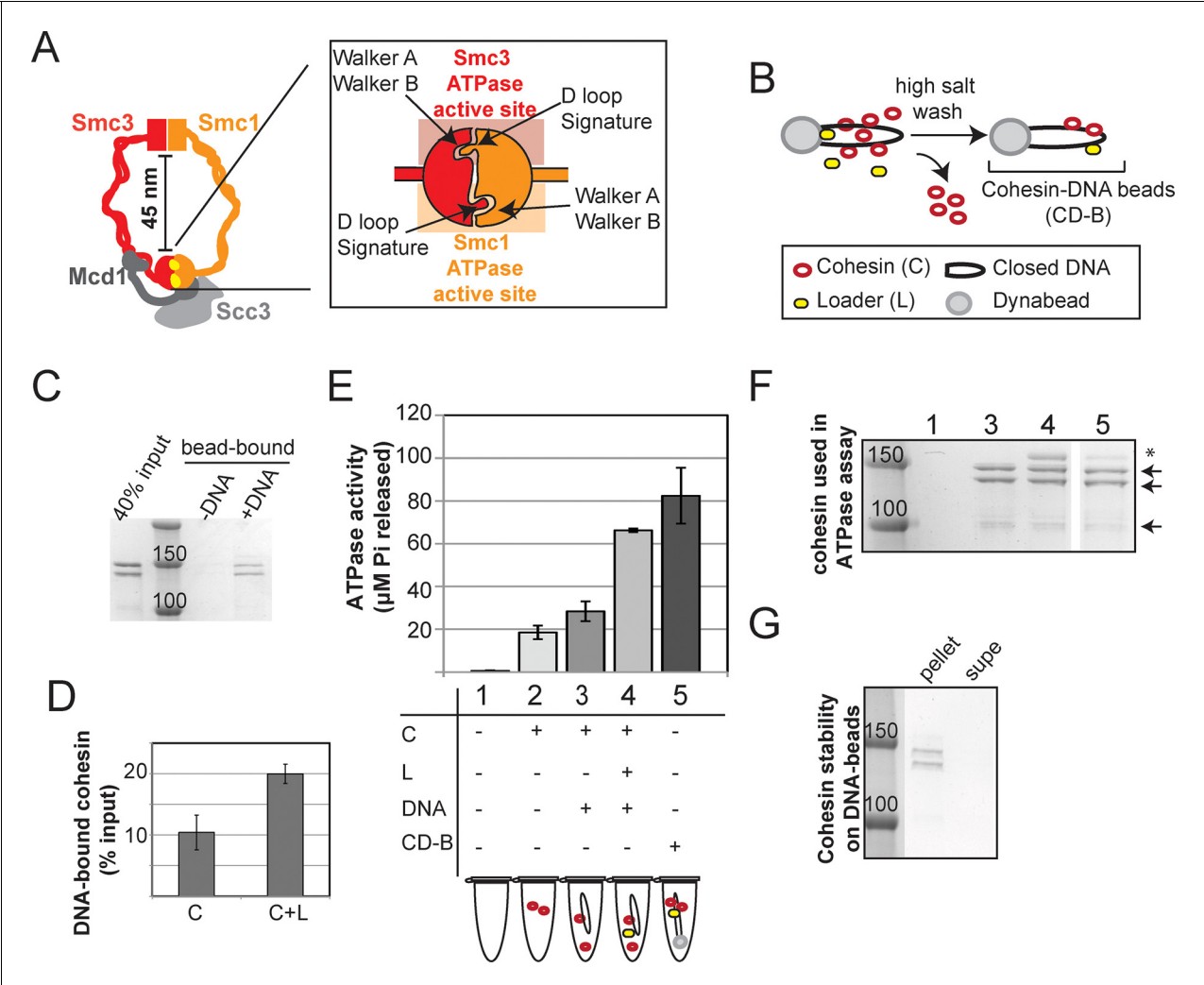

**Figure 1.** Stable DNA binding does not suppress ATPase activity of cohesin. (**A**) Cartoon representation of the cohesin complex. Inset shows a model of the ABC ATPase domain of cohesin complex, based on the X-ray crystal structure of Rad50 in the presence of ATP (*Hopfner et al., 2000*), and of Smc1 head domain in the presence of ATP and Mcd1 C-terminus (*Haering et al., 2004*). This model has not been experimentally verified for the cohesin complex heterodimeric ATPase head domain. (**B**) Schematic showing in vitro assembly of stable cohesin-DNA bead complexes. DNA bearing *CARC1* sequence was attached to dynabeads via biotin-streptavidin interaction at both ends. Cohesin was incubated with bead-bound DNA and loader in buffer containing 25 mM KCl and 25 mM NaCl, then washed in 500 mM KCl to wash off salt-sensitive cohesin. The remaining DNA-bound cohesin (and small amount of loader) is referred to as cohesin-DNA-beads (CD-B). (**C**) Cohesin assembly on DNA-beads. *S. pombe* cohesin and loader complexes were purified from Y4443 and Y4483, respectively. Purified cohesin and loader were incubated with dynabeads-DNA or dynabeads alone for 1 hour at 30°C, then cohesin was washed off as described in B. Cohesin bound DNA-beads (DNA) but failed to bind beads lacking DNA (-). (**D**) Cohesin binding to DNA is stimulated by the loader complex. DNA binding was done as described in B & C, except loader was omitted in one sample. Percent cohesin bound was calculated by quantifying bands on Coomassie-stained SDS-PAGE. Data from two independent experiments, error bars represent standard deviation. (**E**) Effect of stable DNA binding on cohesin ATPase activity. ATPase activity of cohesin alone (2) was compared to cohesin with DNA (3), cohesin in presence of loader complex and DNA (4), and cohesin in stable cohesin-DNA complexes (CD-B, 5). Proteins were incubated in ATPase buffer 2 spiked with hot ATP for 1 hour at 30°C. Released Pi was calculated and plotted as described in Methods. Error bars represent standard deviation from two independent experiments. (**F**) Equal concentrations of cohesin were used in the ATPase reactions. Arrows point to *S. pombe* homologs of cohesin subunits, Smc1, Smc3 (Psm1 and Psm3 in *S. pombe*, ~150 kD), Mcd1 and Scc3 (Rad21 and Psc3 in *S. pombe*, ~100 kD). Asterisks mark subunits of the *S. pombe* homolog of the loader complex, Scc2/Scc4 (Mis4/Ssl3 in *S. pombe*). Due to the lower ability of the loader complex to bind to DNA under these conditions, there is less loader complex present in sample 5 than in sample 4. (**G**) Stably DNA bound cohesin remained bound to DNA-beads throughout the course of the ATPase experiment. Cohesin was incubated with DNA-beads in the presence of loader and ATP as described before and washed in 0.5 M KCl, then resuspended in ATPase buffer. Samples were incubated for 1 hour at 30°C. Supernatant and beads were separated and visualized on SDS-PAGE. Protein gels are representative of at least 2 independent experiments. Bands in some panels were spliced from the same gel for representation purposes. Please see *Figure supplements 1–4* for further characterization of cohesin's ATPase activity while stably bound to DNA.

*Figure 1 continued*

The following figure supplements are available for figure 1:

**Figure supplement 1.** The ATPase activity of cohesin is stimulated by the loader and abolished by a walker-A (K38I) mutation in Psm3 (*S. pombe* Smc3 homolog).

**Figure supplement 2.** Stably DNA-bound cohesin (CD-B) can be eluted off the DNA-beads by a DNase or restriction enzyme (Mnl I) digest.

**Figure supplement 3.** Stably DNA-bound cohesin does not come off the DNA-beads after incubation with competitor DNA.

**Figure supplement 4.** Cohesin stably assembled on DNA in the absence of loader has at least as high ATPase activity as cohesin +DNA.

(*Murayama and Uhlmann, 2014*), cohesin's basal ATPase activity (sample 2) was stimulated about 3-fold by the presence of loader and DNA (sample 4), but only subtly increased by DNA alone (sample 3, *Figure 1E*). Our purified cohesin-DNA complexes on beads (CD-B, sample 5) showed about 4-fold increased ATPase activity compared to the basal activity of equal amounts of cohesin (sample 2, *Figure 1E,F*). Furthermore, the ATPase activity of CD-B was at least as high, if not higher, as the activity of equal concentrations of loader/DNA-stimulated cohesin in solution, where additional rounds of cohesin loading were possible (sample 4 compared to sample 5, *Figure 1E,F*).

The ATPase activity in the fraction containing purified cohesin-DNA complexes might have been derived from DNA-free cohesin that had dissociated from DNA-beads. To test this possibility, a parallel sample of CD-B was treated similarly to the ATPase assay to assess the amount of cohesin that remained bound to DNA through the course of the ATPase experiment. DNA-beads were separated from the supernatant after the 30-minute incubation with ATP and the amount of cohesin in each fraction was visualized by SDS-PAGE. Very little, if any, cohesin was found in the supernatant at the end of the incubation, suggesting that most of the ATPase activity was derived from cohesin bound to the DNA-beads (*Figure 1G*). Even in the presence of excess competitor DNA in solution, cohesin remained stably bound to the DNA-beads (*Figure 1—figure supplement 3*). Thus in vitro, cohesin can have robust ATPase activity while remaining stably bound to DNA.

In our preparations of purified cohesin-DNA complexes, a small amount of the loader complex was also present, due to the ability of the loader complex to bind DNA. Given that the loader complex stimulates cohesin's ATPase activity in the presence of DNA, it was possible that the ATPase activity in our purified cohesin-DNA complexes depended upon the presence of the loader complex. To test this possibility, we repeated these experiments but omitted the loader from the binding reaction. Under these conditions, a smaller percentage of cohesin could be assembled on DNA in a salt-resistant manner (*Figure 1D* and *Figure 1—figure supplement 4*). Cohesin in these purified cohesin-DNA complexes that have no co-purifying loader complex also retained ATPase activity equivalent to that seen in the mixture of cohesin and DNA in solution (*Figure 1—figure supplement 4*). Thus, the persistence of ATPase activity in stable cohesin-DNA complexes was not dependent upon the loader. The presence of cohesin's ATPase activity in the stable cohesin-DNA complex is consistent with a regulatory role for cohesin ATPase in cohesion establishment in vivo after cohesin stably binds to DNA.

## Mutations proximal to the Smc3 ATPase active site bypass the requirement for Eco1 in cohesion establishment

Cohesin exhibited robust ATPase activity after stably binding DNA, which encouraged us to re-examine the in vivo role of cohesin ATPases and Eco1 in cohesin function. Cohesin-DNA complexes in *eco1Δ wpl1Δ* cells are stably bound to DNA but fail to establish cohesion, indicating a post-DNA binding step is required for cohesion. We postulated that this failure to generate cohesion was due to a misregulation of the cohesin ATPase active sites. If so, a subset of mutations altering the Smc1 or Smc3 ATPase active sites might mimic the state of the ATPase upon Eco1 acetylation, and thereby might suppress the cohesion defect of *eco1Δ wpl1Δ* cells.

To identify such suppressors, we exploited the fact that *eco1Δ wpl1Δ* cells remain viable in the absence of cohesion because of an unusual feature of budding yeast cell cycle that gives rise to a

cohesin-independent mechanism of sister chromatid segregation (*Guacci and Koshland, 2012*). The reliance of *eco1Δ wpl1Δ* cells on this cohesin-independent segregation makes them sensitive to the microtubule depolymerizing drug benomyl. A screen for suppressors of the benomyl sensitivity of *eco1Δ wpl1Δ* cells should identify a subset of mutations that restore cohesion and the cohesin-dependent benomyl-resistant mechanism of segregation. Indeed, we recently reported one such benomyl-resistant suppressor allele, *smc3-D1189H* (*Guacci et al., 2015*), that restored cohesion. *smc3-D1189H* is located in the ATPase head domain, near the Smc3 ATPase active site. Due to this restoration of cohesion to *eco1Δ wpl1Δ* cells, we termed this mutation a cohesion activator mutation.

Here, we present two new *SMC1* alleles from our screen, *smc1-D1164E* and *smc1-Y1128C*. Re-introduction of these mutations into the parent *eco1Δ wpl1Δ*background generated benomyl-resistance identical to the initial suppressor *eco1Δ wpl1Δ* isolates and close to the resistance of wild type cells (*Figure 2A*), demonstrating their causal role for this phenotype. The *SMC1* suppressor mutations are even more proximal to the Smc3 ATPase active site than *smc3-D1189H* (*Figure 2B*). *smc1-Y1128C* is adjacent to the signature motif, which is thought to modulate ATP binding, but this residue is not conserved. *smc1-D1164E* alters the invariant aspartate that is part of the conserved D-loop motif found in ABC and SMC ATPases (*Figure 2C*). Studies from other ABC ATPases suggest that this aspartate interacts with the catalytic H loop and Walker A (*Hohl et al., 2012*; *la Rosa and Nelson, 2011*), and mediates communications between the two ATPase active sites within the ATPase head domain (*Furman et al., 2013*). Taken together, it is likely that these suppressor mutations modulate the cohesin ATPase to promote cohesion.

To begin to test this hypothesis, we first asked whether *smc1-D1164E* or *smc1-Y1128C* were also cohesion activator mutations, i.e. suppressed the severe cohesion defect of *eco1Δ wpl1Δ* cells. Haploid wild type cells and *eco1Δ wpl1Δ* parent cells containing *SMC1*, *smc1-D1164E* or *smc1-Y1128C* alleles were arrested in G1, then released into media containing nocodazole to arrest them in M phase (*Figure 3A*). To assay for cohesion, these cells contained a LacO array integrated at a *CEN*-proximal locus (*TRP1*), and a GFP-LacI fusion protein that bound to the array. In this assay, sister chromatids that have cohesion generate a single GFP spot, whereas chromatids lacking cohesion generate 2 GFP spots (*Figure 3A*). Wild type cells had robust cohesion as shown by few M phase cells with two GFP spots (*Figure 3B* and *Figure 3—figure supplement 1A*). *eco1Δ wpl1Δ* cells exhibited a severe cohesion defect (more than 60% of cells had two GFP spots), which was reduced by half by the *smc1-D1164E* suppressor (*Figure 3B* and *Figure 3—figure supplement 1A*). The *smc1-Y1128C* suppressor also reduced the *eco1Δ wpl1Δ* cohesion defect, but less effectively (*Figure 3—figure supplement 2*). Thus, both of these new suppressor mutations, like *smc3-D1189H*, suppressed significantly but not completely the cohesion defect of *eco1Δ wpl1Δ* cells. The increased *CEN*-proximal cohesion generated by *smc1-D1164E* or *smc1-Y1128C* was likely responsible for the increased benomyl resistance in the *eco1Δ wpl1Δ* background. Subsequent studies focused on *smc1-D1164E*, because its alters a critical and absolutely conserved D-loop aspartate, and it more robustly suppresses the cohesion defect of *eco1Δ wpl1Δ* cells.

To better understand the impact of the *smc1-D1164E* suppressor on cohesion, we monitored cohesion at the *CEN*-proximal *TRP4* locus at 20 minute intervals as cells progressed from G1 to M (*Figure 3—figure supplement 1B*). As expected, wild type cells showed robust cohesion where very few cells had separated sisters, and *eco1Δ wpl1Δ* cells showed sister chromatid separation starting in S phase (*Figure 3C*), consistent with a cohesion establishment defect that was shown previously for this strain (*Guacci and Koshland, 2012*; *Guacci et al., 2015*). The small amount of cohesion loss in *smc1-D1164E eco1Δ wpl1Δ* cells also began during S phase, indicating that the residual cohesion defect in these cells resulted from a failure in cohesion establishment. However, these cells displayed fewer separated sisters throughout the time course compared to *eco1Δ wpl1Δ* cells. These results indicated that (1) *smc1-D1164E* promotes cohesion establishment in the *eco1Δ wpl1Δ*, albeit incompletely, and (2) once cohesion is formed in *smc1-D1164E eco1Δ wpl1Δ* cells, it is maintained. *smc1-D1164E* also partially restored cohesion establishment in *eco1Δ wpl1Δ* cells at the *CEN*-distal locus, *LYS4* (*Figure 3—figure supplement 3*). We conclude that *smc1-D1164E* promotes cohesion establishment in the *eco1Δ wpl1Δ*, albeit incompletely.

Since cohesin is already stably bound to DNA in *eco1Δ wpl1Δ* cells, the restoration of cohesion in *smc1-D1164E eco1Δ wpl1Δ* cells should occur without altering the DNA binding of cohesin. To test this prediction, we performed chromosome immunoprecipitation (ChIP) to compare cohesin binding

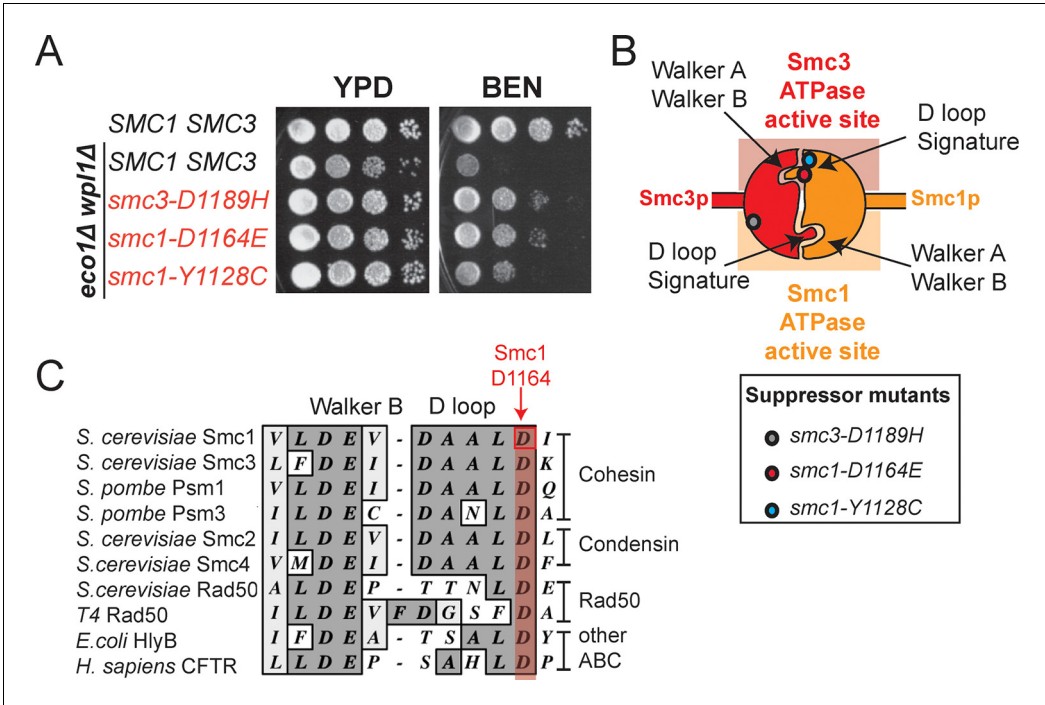

**Figure 2.** smc1-D1164E and smc1-Y1128C mutations suppress the benomyl sensitivity of eco1△ wpl1△ cells and are part of the Smc3 ATPase active site. (**A**) Assessing effects of *smc1-D1164E* and *smc1-Y1128C* on *eco1△ wpl1△* benomyl sensitivity. Haploid wild-type cells (VG3460-2A), and *eco1△ wpl1△* cells alone (VG3502-1A) or containing *smc1-D1164E* (VG3574-5A), *smc1-Y1128C* (VG3576-1C) or *smc3-D1189H* (VG3547-3B) were grown to saturation in YPD at 23°C, plated as 10-fold serial dilutions on YPD alone, or containing benomyl at 12.5 µg/mL (BEN) then incubated at 23°C for 3 days. (**B**) Cartoon depicting the Smc1 and Smc3 ATPase active sites along with the position of three suppressor mutations shown in A. All three suppressor mutations are in the vicinity of the Smc3 ATPase active site. Note that the Smc1 D-loop and signature motifs form part of the Smc3 ATPase active site. (**C**) The conservation of residues around the D-loop in distant ABC ATPases.

to chromosomes in *eco1△ wpl1△* cells containing *SMC1* or *smc1-D1164E* allele. Cohesin subunits colocalize on chromosomes, making the analysis of any cohesin subunit a surrogate for cohesin binding (*Glynn et al., 2004*; *Heidinger-Pauli et al., 2010a*; *Lengronne et al., 2004*). Here we used anti-Mcd1 antibodies as a means to assess cohesin binding to chromosomes. Cohesin binding at CARs has been shown to be reduced 2-3 fold in *eco1△ wpl1△* cells as compared to wild type cells (*Guacci et al., 2015*; *Sutani et al., 2009*). In *eco1△ wpl1△* cells containing either *SMC1* or *smc1-D1164E* allele, there was little or no difference in Mcd1 binding at the centromere-proximal *CARC1* or centromere-distal *CARL1* (*Figure 3—figure supplement 4*). This similarity in chromosomal binding indicates that cohesion restoration in *eco1△ wpl1△* cells by the *smc1-D1164E* allele was not due to an altered level of cohesin binding to chromosomes, but instead was due to a change in cohesin function at a post-DNA binding step.

## *smc1-D1164E* has constitutive but reduced cohesion independent of Eco1 and Wpl1

The restoration of cohesion in *smc1-D1164E eco1△ wpl1△* cells indicated that the *smc1-D1164E* could bypass one or more aspects of cohesion regulation normally imposed by Eco1 and Wpl1. To understand better how *smc1-D1164E,* and by inference the Smc3 ATPase active site, interfaced with cohesin regulators, we examined its impact on the phenotypes of *wpl1△* and *eco1△*single mutants. In a *wpl1△* background, *SMC1*and *smc1-D1164E* alleles had similar viability and the same moderate cohesion defect characteristic of *wpl1△* mutants (*Figure 3—figure supplement 5*). The *smc1-D1164E* allele suppressed the essential *ECO1* function, as *smc1-D1164E eco1△* cells showed good viability (*Figure 3—figure supplement 6*). *smc1-D1164E eco1△* cells showed a 35% cohesion defect

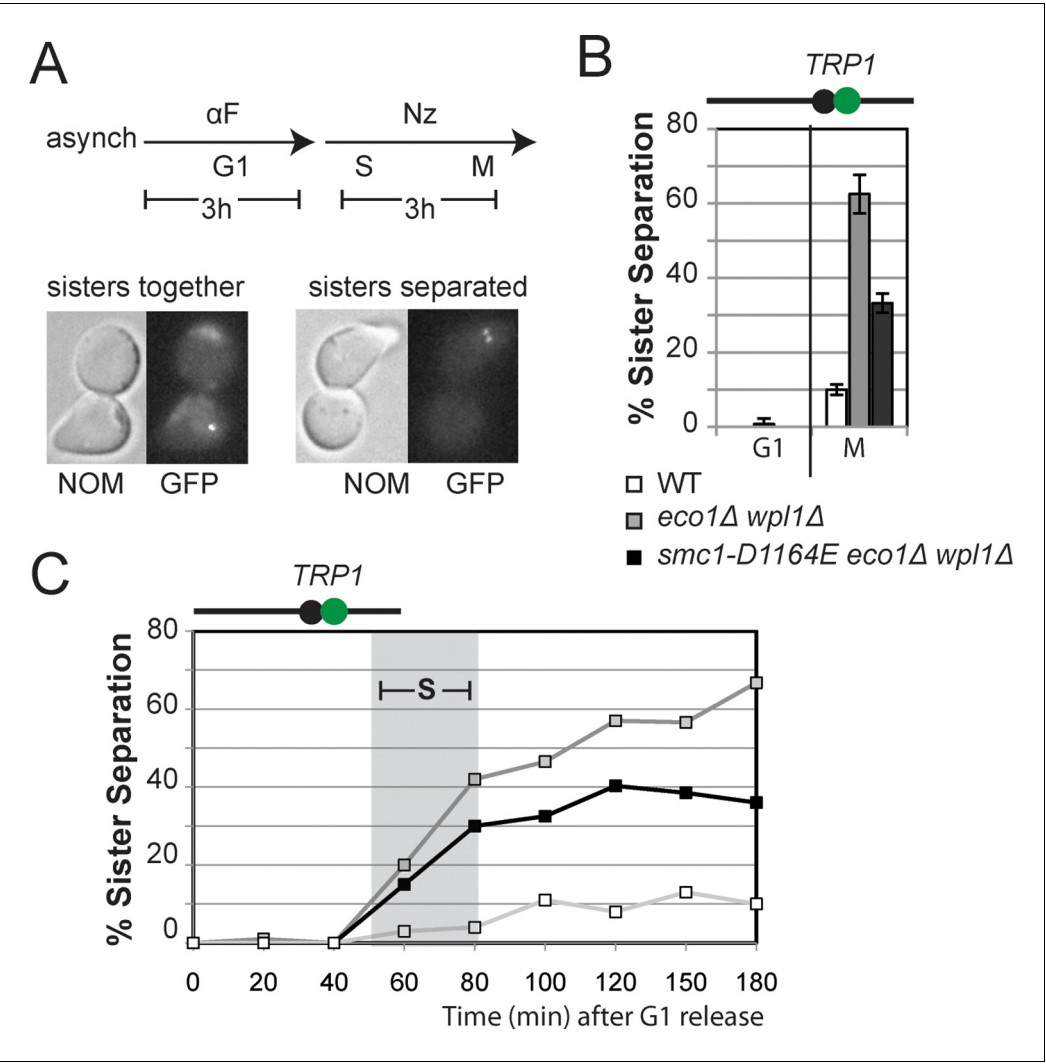

**Figure 3.** *smc1-D1164E* allele restores cohesion in the absence of both Eco1 and Wpl1. (**A**) Regimen used to assess sister chromatid cohesion in cells. Mid-log phase cultures of asynchronously growing cells at 23°C were arrested in G1 with alpha factor for 3 hours, then released into media containing nocodazole for 3 hours to arrest cells in M phase. Representative images of M phase arrested cells are shown with cells being visualized by Nomarski (NOM) and cohesion (GFP), which marks a *CEN*-proximal *TRP1* locus. Left side shows a cell where cohesion exists (one GFP spot.) Right side shows a cell where sisters have separated (2 GFP spots). (**B,C**) *smc1-D1164E* partially restored cohesion in *eco1△ wpl1△* cells at the *CEN*-proximal *TRP1* locus. Haploid wild type (WT, VG3460-2A), *eco1△ wpl1△* (VG3502-2A) and *smc1-D1164E eco1△ wpl1△* (VG3574-5A) cells were released from G1 and arrested in M phase using nocodazole as described in A. The percentage of cells with two GFP spots was plotted. (**B**) Cohesion loss at a *CEN*-proximal locus (*TRP1*) in M phase arrested cells. DNA content of these cells is shown in *Figure 3—figure supplement 1*, panel A. (**C**) Time course to assess kinetics of cohesion loss at a *CEN*-proximal locus (*TRP1*). Cell aliquots were fixed in G1 and at 20-minute intervals after release. Grey box shows S phase (based on DNA content shown in *Figure 3—figure supplement 1*, panel B). Please see Figure supplements 1–6 for further characterization of cohesin activator mutants.

The following figure supplements are available for figure 3:

**Figure supplement 1.** DNA content analysis of cells from *Figure 3B(A)* and *Figure 3C(B)*.

**Figure supplement 2.** Cohesion loss in *smc1-Y1128C* at *CEN*-proximal (*TRP1*) locus.

**Figure supplement 3.** Cohesion loss at a *CEN*-distal locus (*LYS4*) in *smc1-D1164E* cells.

*Figure 3 continued on next page*

*Figure 3 continued*

**Figure supplement 4.** Effect of the *smc1-D1164E* on cohesin (Mcd1) binding in M phase arrested *eco1Δ wpl1Δ* cells.

**Figure supplement 5.** The Smc3 ATPase active site D-loop cohesion activator mutation *smc1-D1164E* cannot suppress a *wpl1Δ*.

**Figure supplement 6.** The Smc3 ATPase active site D-loop cohesion activator mutation smc1-D1164E partially suppresses the requirement for Eco1.

(*Figure 3—figure supplement 6*), which was greatly reduced compared to 65-70% cohesion defect that was observed previously for *eco1* mutant cells or cells lacking both Eco1 and Wpl1 (*Guacci et al., 2015*; *Sutani et al., 2009*). Taken together, these results suggest that *smc1-D1164E* restored cohesion by phenocopying both Eco1's antagonism of Wpl1 and its ability to promote stably DNA-bound cohesin to tether sister chromatids.

The partial restoration of cohesion in *smc1-D1164E eco1Δ wpl1Δ* cells could reflect the inability of *smc1-D1164E* to fully compensate for the loss of Eco1 or Wpl1. Alternatively, it might reflect an inherent cost of uncoupling cohesin from its regulators. To differentiate between these possibilities, we characterized the *smc1-D1164E* allele when both *ECO1* and *WPL1* were present. In plasmid shuffle assays (Materials and methods), the viability of cells containing the *smc1-D1164E* or *SMC1* alleles was indistinguishable (*Figure 4—figure supplement 1*). However, otherwise wild type cells carrying the *smc1-D1164E* allele had a moderate cohesion defect at both *CEN*-proximal and *CEN*-distal loci, which arose around the time of replication (*Figure 4—figure supplement 2*). This defect was similar to that observed for the *smc1-D1164E eco1Δ wpl1Δ* cells. Thus, cohesin in *smc1-D1164E* cells had an inherent moderate cohesion establishment defect independent of cohesin regulators. Taken together, our results suggest that the *smc1-D1164E* allele was constitutively cohesive as it uncoupled cohesin function from its regulators, which reduced the efficiency of cohesion generation.

## Analogous D-loop mutations in Smc1 and Smc3 ATPase active sites differentially affect cohesin's DNA-binding and DNA-tethering activities

Our two new cohesion activator mutations mapped to residues intimately involved in the Smc3 ATPase active site, and the previously characterized third cohesion activator was in close proximity (*Guacci et al., 2015*). This observation suggested that the Smc3 but not the Smc1 ATPase active site might play a specialized role in activating stably DNA-bound cohesin to tether sister chromatids.

To test more directly whether the two active sites played distinct roles in cohesin regulation, we compared the phenotypes of *smc1-D1164E* to *smc3-D1161E*, the analogous glutamate substitution of the D-loop aspartate of the Smc1 ATPase active site (*Figure 4A*). We used Auxin Inducible Degron (AID) system to obtain conditional alleles of *SMC1* and *SMC3*, *SMC1-AID* and *SMC3-AID,* respectively, which can be degraded upon the addition of auxin (*Figure 4—figure supplement 3*). In strains containing either *SMC1-AID* or *SMC3-AID* alleles as sole source of Smc1 or Smc3, we integrated our test alleles, *smc1-D1164E* and *smc3-D1161E*, respectively. The *smc1-D1164E* allele robustly suppressed the inviability associated with depletion of Smc1-AID, whereas *smc3-D1161E* could not support viability when Smc3-AID was depleted (*Figures 4B,C*). Similar results in viability were seen when these alleles were tested by plasmid shuffle assays (*Figure 4—figure supplement 1 & 4*). This difference in viability between the *smc1-D1164E* and *smc3-D1161E* alleles suggests that Smc1 and Smc3 ATPase active sites have distinct functions.

We next assessed cohesion at *CEN*-proximal and *CEN*-distal loci in *smc1-D1164E SMC1-AID* and *smc3-D1161E SMC3-AID* strains under conditions in which AID tagged subunits remain depleted from G1 through M phase (*Figure 4D*). As expected, *SMC1-AID* and *SMC3-AID* cells showed severe cohesion defects that could be rescued by *SMC1* and *SMC3*, respectively (*Figure 4E* and *Figure 4—figure supplement 5A*). *smc1-D1164E SMC1-AID* cells had only a modest cohesion defect, in line with our previous results for the *smc1-D1164E* in an otherwise wild type background. In contrast, in the presence of auxin, the *smc3-D1161E SMC3-AID* cells were severely compromised for cohesion, at levels comparable to cells that only have the *SMC3-AID* allele (~90% separated sisters, *Figure 4F*

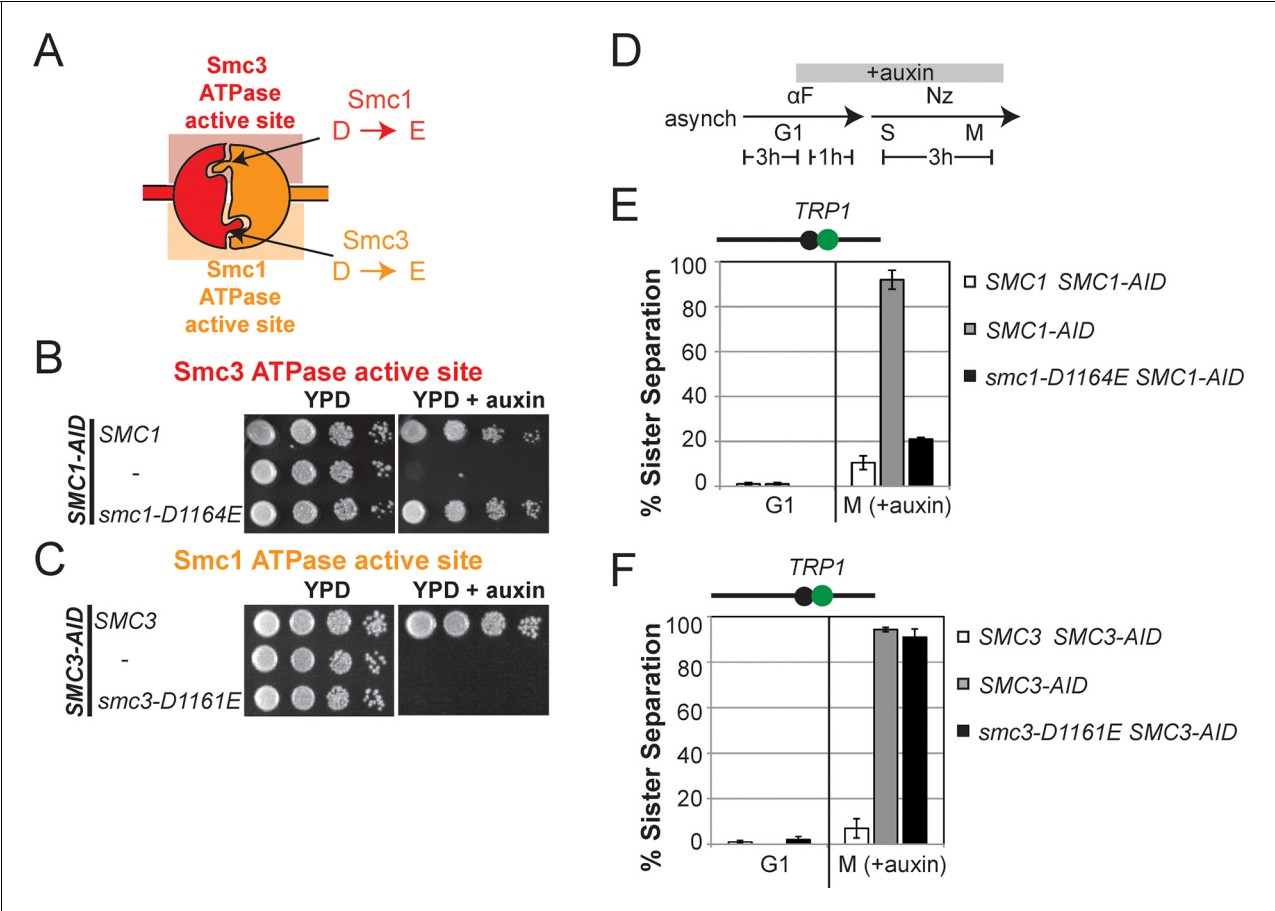

**Figure 4.** D-loop mutations in Smc1 and Smc3 ATPase active sites reveal that the two sites are not functionally equivalent. (**A**) Cartoon representation of cohesin's ATPases. The Smc1-encoded D-loop is part of the Smc3 ATPase active site and the Smc3-encoded D-loop is part of the Smc1 ATPase active site. (**B**) Assessing whether the Smc3 ATPase active site D-loop mutant *smc1-D1164E* promotes cell viability. Haploid *SMC1 SMC1-AID* (VG3764-3A), *SMC1-AID* (VG3711-5D), and *smc1-D1164E SMC1-AID* (VG3765-3D) cells were grown and plated on as described in *Figure 2A* onto YPD alone or YPD + auxin and incubated 3 days at 23°C. *SMC1-AID* was depleted in media containing auxin, which allows assessment of whether *smc1-D1164E* promotes viability. (**C**) Assessing whether the Smc1 ATPase active site D-loop mutant *smc3-D1161E* promotes cell viability. Haploid *SMC3 SMC3-AID* (VG3771-10C), *SMC3-AID* (VG3651-3D), and *smc3-D1161E SMC3-AID* (VG3773-16D) were grown and plated dilution as described in C. (**D**) Regimen used to assess sister chromatid cohesion in cells containing AID tagged proteins. Asynchronous cells were arrested in G1, depleted for AID tagged proteins by the addition of auxin, then released from G1 and arrested in M phase in the presence of auxin. (**E,F**) Cohesion loss at the *CEN*-proximal locus (*TRP1*) in M phase cells depleted for AID tagged proteins. *SMC1-AID* or *SMC3-AID* was depleted in strains from G1 through M phase as described in D. The percentage of cells with two GFP spots (sister separation) is plotted. (**E**) *smc1-D1164E* promotes cohesion in *SMC1-AID* depleted cells. Haploid *SMC1 SMC1-AID* (VG3794-2E), *SMC1-AID* (VG3711-5D) and *smc1-D1164E SMC1-AID* (VG3795-2C) assayed for cohesion. DNA content analysis of these cells can be seen in *Figure 4—figure supplement 4*. (**F**) *smc3-D1161E* fails to promote cohesion in to promote cohesion *SMC1-AID* depleted cells. Haploid *SMC3 SMC3-AID* (VG3797-1A), *SMC3-AID* (VG3651-3D) and *smc3-D1161E SMC3-AID* (VG3799-3C) strains assayed for cohesion. DNA content analysis of these cells can be seen in *Figure 4—figure supplement 4*. Please see Figure supplements 1–6 for further characterization of *smc1-D1164E* and *smc3-D1161E* alleles.

The following figure supplements are available for figure 4:

**Figure supplement 1.** Viability of *smc1-D1164E* allele as sole source of *SMC1*.

**Figure supplement 2.** Time course assessing kinetics of cohesion loss in *SMC1* and *smc1-D1164E* cells.

**Figure supplement 3.** Assessing *SMC-AID* depletion and Mcd1 levels after depletion.

**Figure supplement 4.** Inviability of *smc3-D1161E* allele as sole source of *SMC3*.

**Figure supplement 5.** DNA content analysis of cells used in *Figure 4E,F*.

*Figure 4 continued on next page*

*Figure 4 continued*

**Figure supplement 6.** Cohesion assay of *smc1-D1164E* and *smc3-D1161E* at a *CEN*-distal locus.

and *Figure 4—figure supplement 5B*). Similar results were obtained when cohesion was assessed at the *CEN*-distal locus (*Figure 4—figure supplement 6*). Thus, *smc1-D1164E* and *smc3-D1161E* differ in their ability to promote viability and cohesion.

To understand the molecular basis for this difference in cohesion, we performed ChIP on this panel of strains to assess cohesin binding to chromosomes under conditions in which the AID-tagged proteins were depleted. The level of cohesin binding to chromosomes in *smc1-D1164E SMC1-AID* cells was reduced about twofold compared to *SMC1 SMC1-AID* (*Figure 5A* and *Figure 5—figure supplement 1*). In contrast, cohesin binding to chromosomes in *smc3-D1161E SMC3-AID* cells was essentially at the background levels observed in *SMC3-AID*-only cells (*Figure 5B* and *Figure 5—figure supplement 1*). Taken together, these data suggest that the Smc1 ATPase active site regulates cohesin ATPase to modulate cohesin's DNA binding, whereas the Smc3 active site regulates cohesin ATPase to modulate cohesin-mediated DNA tethering after its stable binding to DNA.

## The D-loop mutants of Smc1 and Smc3 ATPase active sites uncouple the level of ATPase activity from DNA binding and tethering

Precedent that the D-loop mutations might impact ATPase activity come from studies of the homo-dimeric SMC protein Rad50. The mutation of the D-loop aspartate in Rad50 to an alanine (Rad50$^{DA}$) dramatically reduced its ATPase activity (*la Rosa and Nelson, 2011*) (*Figure 5—figure supplement 2*). Furthermore, the substitution of the D-loop aspartate with a glutamate (Rad50$^{DE}$) led to a 3-fold reduction in ATPase activity (*Figure 5—figure supplement 2*).

To address how the cohesin ATPase activity would be affected by these substitutions, we purified the *S. pombe* cohesin complex with the analogous mutations (*Murayama and Uhlmann, 2014*) and assayed the ATPase activity (a sum of the ATP hydrolysis by Smc1 and Smc3 active sites) for equivalent amounts of wild type and mutant cohesin complexes. The mutation in the *S. pombe* homolog of cohesin analogous to *smc1-D1164E* (*Sp*Smc1$^{DE}$), which was competent for cohesion and viability in *S. cerevisiae*, reduced the ATPase activity of cohesin to levels close to the Smc3 walker A mutant (*Sp*Smc3$^{WA}$, *Figure 5C*). In contrast, the D-loop-E mutation in the Smc1 ATPase active site (*Sp*Smc3$^{DE}$), analogous to the *smc3-D1161E* allele that abolished chromosomal association of cohesin, led to a subtler reduction in cohesin's total ATPase activity. These results suggested that the substitution of the D-loop aspartate with a glutamate in Smc3 and Smc1 ATPase active sites affect cohesin in unique ways that uncouple the level of ATPase activity from DNA tethering and DNA binding.

The novel and distinct phenotypes of *smc1-D1164E* and *smc3-D1161E* might result from a subtle change that alters rather than abrogates D-loop function, given the chemical similarity of aspartate and glutamate. To test whether these unusual phenotypes persisted with more radical substitutions of the D-loop aspartate, we changed it to alanine in the Smc1 or Smc3 ATPase active sites. We introduced these *smc1-D1164A* or *smc3-D1161A* alleles into strains bearing *SMC1-AID* or *SMC3-AID* alleles, respectively (Figure 6A). We then characterized these alleles under conditions in which the AID-tagged proteins were degraded (*Figure 6—figure supplement 1*). Neither *smc1-D1164A* nor *smc3-D1161A* allele was able to sustain viability on auxin plates (*Figures 6B,C*). Similar results in viability were seen using plasmid shuffle assays (*Figure 6—figure supplement 2*). Moreover, the *smc1-D1164A* or *smc3-D1161A* cells were both severely compromised for cohesion, although the cohesion defect in *smc3-D1161A* was more severe (*Figures 6D,E*, *Figure 6—figure supplements 3 and 4*). ChIP using anti-Mcd1 antibodies showed that *smc1-D1164A* had 7-10 fold less cohesin binding than wild type, whereas *smc3-D1161A* reduced cohesin binding to chromosomes to background levels seen with the *SMC3-AID* alone (*Figures 7A,B*). Similar results were obtained when tagged smc1-D1164A and smc3-D1161A proteins were assessed by ChIP (*Figure 7—figure supplement 1*). Therefore, the glutamate substitution in the D-loop of the Smc3 ATPase active site is unique in its

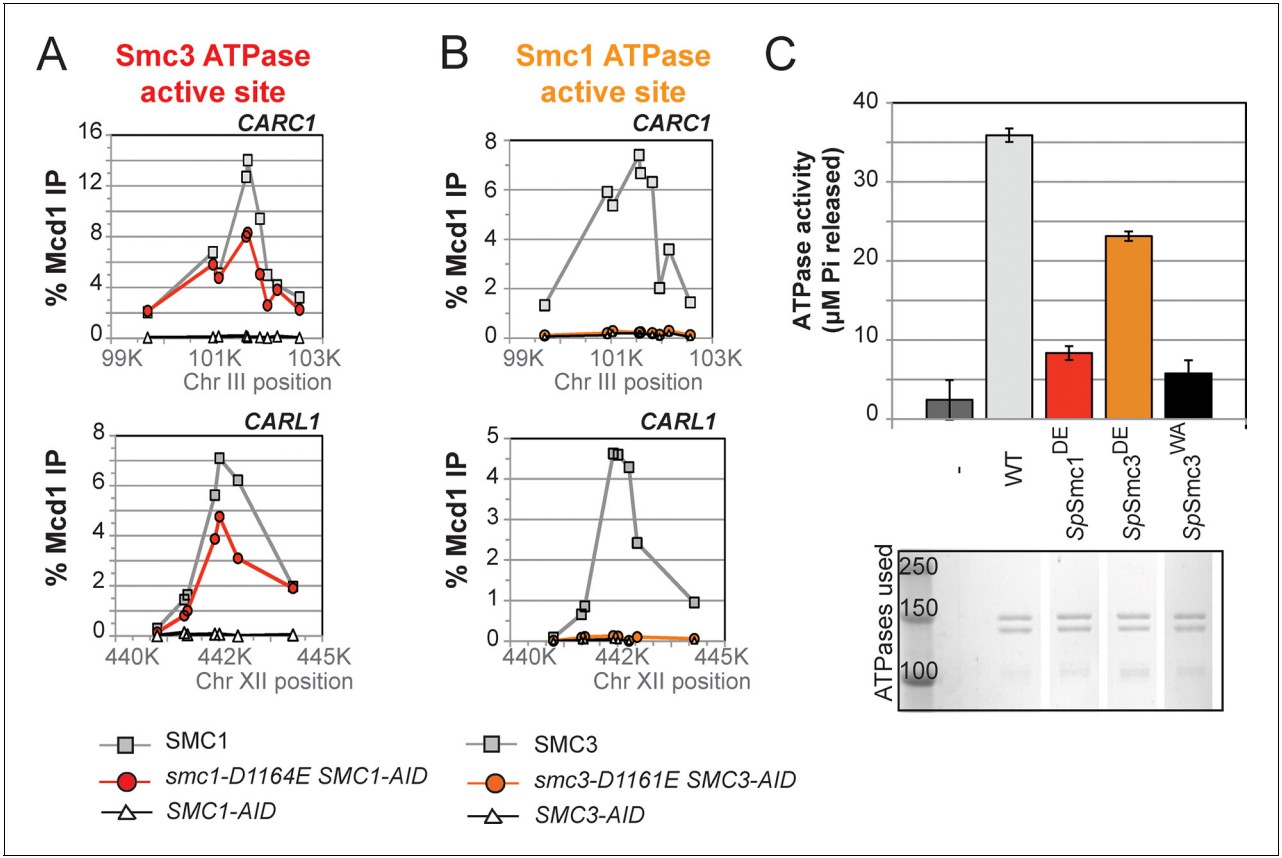

**Figure 5.** D-to-E mutations in Smc1 and Smc3 ATPase active sites uncouple the level of ATPase activity from chromosome binding. (A,B) ChIP of Mcd1 in M phase cells depleted for AID tagged proteins. G1 cells were depleted for AID tagged proteins then released under depletion condition and arrested in M phase as described in *Figure 4D*. M phase cells were fixed and processed for ChIP using Mcd1 antibodies, and the% Mcd1 binding plotted as described in *Figure 3—figure supplement 4*. (A) ChIP of Mcd1 in *smc1-D1164E* cells at centromere-proximal *CARC1* (top panel) and centromere-distal *CARL1* (bottom panel). Haploid M phase cells from *Figure 4B*, expressing Smc1³ᶠᴸᴬᴳ Smc1ᴬᴵᴰ (*SMC1*; light grey line, light grey squares), Smc1-D1164E³ᶠᴸᴬᴳ Smc1ᴬᴵᴰ (*smc1-D1164E SMC1-AID*; red line, red circles), and Smc1ᴬᴵᴰ alone (*SMC1-AID*; black line, open triangles) were used for ChIP under conditions in which AID-tagged proteins were degraded. (B) ChIP of Mcd1 in M phase in *smc3-D1161E* cells at centromere-proximal *CARC1* (top panel) and centromere-distal *CARL1* (bottom panel). Haploid M phase cells from *Figure 4C* expressing Smc3⁶ᴴᴬ Smc3ᴬᴵᴰ (*SMC3*; light grey line, light grey squares), Smc3-D1161E⁶ᴴᴬ Smc3ᴬᴵᴰ (*smc3-D1161E SMC3-AID*; orange line, orange circles), and Smc3ᴬᴵᴰ alone (*SMC3-AID*; black line, open triangles) were used for ChIP under conditions in which AID-tagged proteins were degraded. (C) ATPase activity of purified *S. pombe* cohesin bearing D-loop mutations Psm1-D1167E or Psm3-D1132E, analogous to *smc1-D1164E* or *smc3-D1161E*, respectively. Same amount of cohesin was used in ATPase experiments (lower panel). Cohesin complexes were purified from cells overexpressing wild type *S. pombe* cohesin (WT) or *S. pombe* cohesin with mutations analogous with Smc3 ATPase active site D-loop-E mutation (*Sp*Smc1ᴰᴱ) or Smc1 ATPase active site D-loop mutation (*Sp*Smc3ᴰᴱ). ATPase assays were carried out in ATPase buffer 1 for 2 hours at 30°C. Cohesin with a K-to-I mutation in the Walker A motif of Smc3 ATPase active site (*Sp*Smc3ᵂᴬ, Psm3 K38I in *S. pombe*) abolished most, if not all, ATPase activity. Coomassie-stained protein bands were spliced from the same gel for representation purposes. Please see figure supplement 1 for further characterization of chromosomal association of cohesin in *smc1-D1164E* or *smc3-D1161E*. Figure supplement 2 shows ATPase activity of Rad50 protein when the D-loop residue is mutated to an E or an A.

The following figure supplements are available for figure 5:

**Figure supplement 1.** Chromosomal binding of cohesin in *smc1-D1164E* and *smc3-D1161E* mutants assayed by antibodies against tagged Smc1 and Smc3 subunits.

**Figure supplement 2.** ATPase activity of wild type and D-loop mutant Rad50 homodimer.

ability to support cohesion suggesting it alters rather than abolishes the in vivo function of the Smc3 ATPase active site.

Finally, we asked whether analogous aspartate-to-alanine substitution mutations altered the ATPase activity of purified *S. pombe* cohesin. Both mutations (*Sp*Smc1ᴰᴬ and *Sp*Smc3ᴰᴬ) reduced

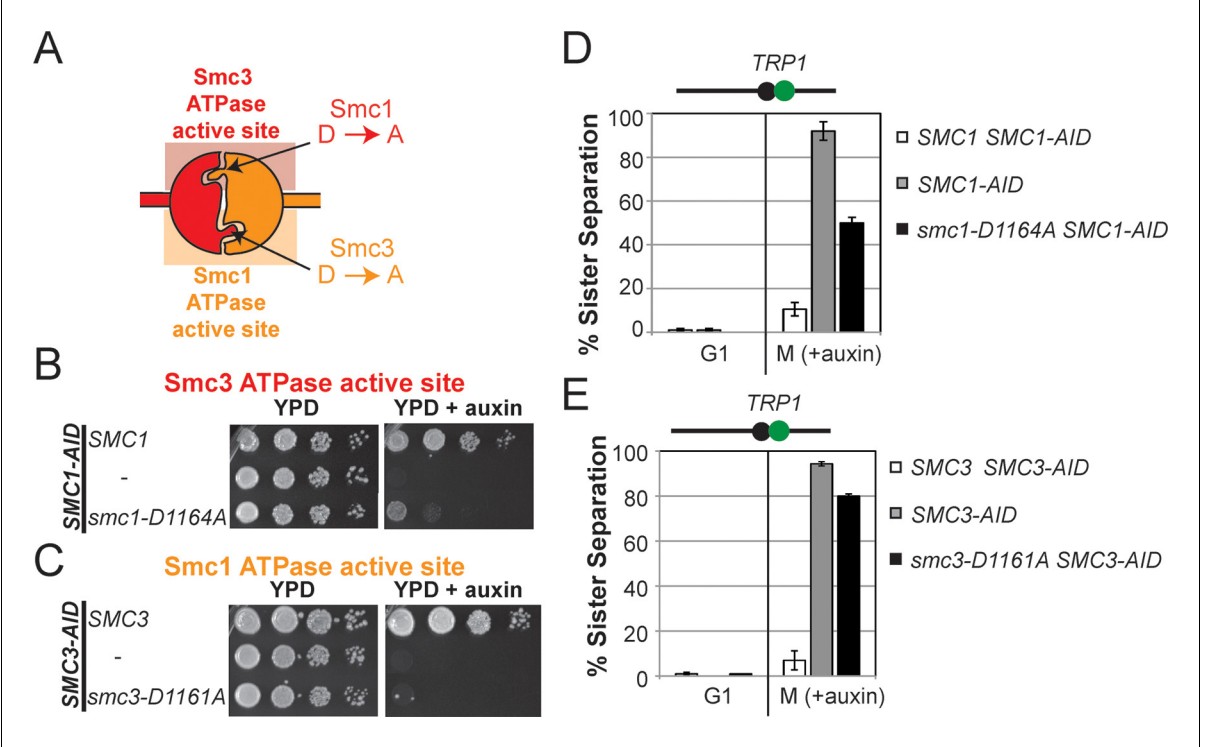

**Figure 6.** D-loop-A (DA) mutations perturb cohesin function more severely than D-loop D to E (DE) mutations (**A**) Cartoon representation of the D-to-A substitution mutants in the Smc3 ATPase active site (*smc1-D1164A*) and Smc1 ATPase active site (*smc3-D1161A*). (**B,C**) Assessing whether the DA D-loop mutants can support viability. (**B**) Smc3 ATPase active site D-loop-A mutant, *smc1-D1164A*, failed to promote viability. *SMC1-AID smc1-D1164A* (VG3766-3C) cells were grown and dilution plated as described in *Figure 4B*. *SMC1-AID SMC1*(VG3764-3A), *SMC1-AID* (VG3311-5D) cells were re-plated here for comparison. The *smc1-D1164A SMC1-AID* was moved from a different region of this same plate for clarity of presentation. (**C**) Smc1 ATPase active site D-loop-A mutant, *smc3-D1161A*, failed to promote viability. *SMC3-AID smc3-D1161A* (VG3772-13A) cells were grown and dilution plated. *SMC3-AID SMC3*(VG3726-6A) and *SMC3-AID* (VG3711-5D) cells were re-plated here for comparison. The *smc3-D1161A SMC3-AID* was moved from a different region of this same plate for clarity of presentation. (**D,E**) Cohesion loss at the *CEN*-proximal locus (*TRP1*) in M phase cells depleted for AID tagged proteins from G1 through M phase as described in *Figure 4D*. The percentage of cells with two GFP spots (sister separation) was plotted. (**D**) Cohesion loss in *smc1-D1164A* cells at the *CEN*-proximal *TRP1* locus. Haploid strains *SMC1 SMC1-AID* (VG3794-2E), *SMC1-AID* (VG3711-5D) and *smc1-D1164A SMC1-AID* (VG3796-1F) assayed for cohesion. (**E**) Cohesion loss in *smc3-D1161A* cells at the *CEN*-proximal *TRP1* locus. Haploid strains *SMC3 SMC3-AID* (VG3797-1A), *SMC3-AID* (VG3651-3D) *smc3-D1161A SMC3-AID* (VG3798-2B) assayed for cohesion. Note: *smc1-D1164A SMC1-AID* and *smc3-D1161A SMC3-AID* cells were analyzed in the same experiments as *Figure 4D,E*, respectively. *smc-DA* data was omitted from *Figure 4* but presented here with controls from those experiments for clarity of presentation. DNA content analysis of cells in *Figure 6D,E* is shown in *Figure 6— figure supplement 3*. Please see Figure supplements 1–6 for further characterization of *smc1-D1164A* and *smc3-D1161A* alleles.

The following figure supplements are available for figure 6:

**Figure supplement 1.** Assessing *SMC-AID* depletion and Mcd1 levels after depletion in DA strains.

**Figure supplement 2.** Characterization of *smc1-D1164A* and *smc3-D1161A* mutants.

**Figure supplement 3.** DNA content analysis of cells in *Figure 6D(A)* and *Figure 6E(B)*.

**Figure supplement 4.** Cohesion of *smc1-D1164A* and *smc3-D1161A* at the *CEN*-distal locus.

the ATPase activity of cohesin to levels comparable to the walker A mutation (*Sp*Smc3[WA], *Figure 7C*). The similar severe defects in ATPase activities for the *S. pombe* Smc1[DE] and Smc1[DA] cohesin complexes was striking given the dramatic differences in the analogous complexes to promote viability, cohesion and cohesin binding to DNA in *S. cerevisiae* (*Figures 4–7*). This functional difference reinforces the conclusion that the cohesin's DNA binding and DNA tethering activities can be uncoupled from the level of cohesin ATPase activity.

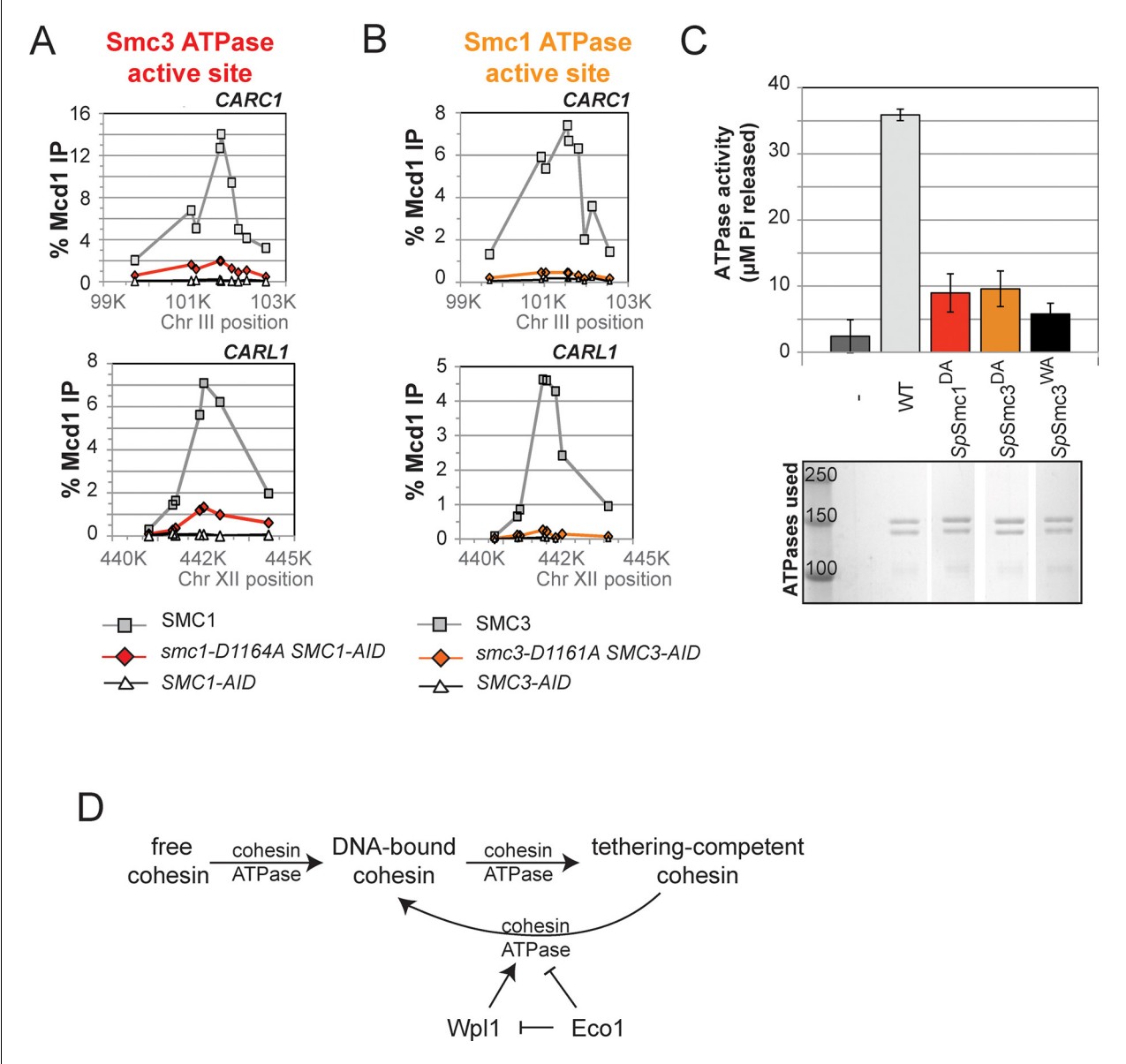

**Figure 7.** DA mutations in Smc1 and Smc3 ATPase active sites abolish chromosome binding and ATPase activity. (**A**) ChIP of Mcd1 in M phase *smc1-D1164A* cells grown in auxin-containing media. Haploid M phase cells from *Figure 6B* were fixed and processed for ChIP using Mcd1 antibodies.% Mcd1 binding plotted as described in *Figure 3—figure supplement 4*. Mcd1 ChIP at centromere-proximal *CARC1* (top panel) and centromere-distal *CARL1* (bottom panel). (**B**) ChIP of Mcd1 in M phase *smc3-D1161A* cells grown in auxin-containing media. Haploid M phase cells from *Figure 6C* were fixed and processed for ChIP. Mcd1 ChIP at centromere-proximal *CARC1* (top panel) and centromere-distal *CARL1* (bottom panel). (**C**) ATPase activity of purified cohesin complexes from *S. pombe* bearing D-loop-A mutations in Smc3 and Smc1 ATPase sctive sites. Psm1 D1167A (*Sp*Smc1$^{DA}$) and Psm3 D1132A (*Sp*Smc1$^{DA}$), analogous to *smc1-D1164A* and *smc3-D1161A*, respectively, were purified from overexpression strains listed in *Supplementary file 1*. Same amount of cohesin was used in the ATPase experiments (lower panel). ATPase assays were carried out in ATPase buffer 1 for 2 hours at 30°C. ATPase activity of wild type (WT) and Walker A-mutant (*Sp*Smc3$^{WA}$) cohesin was represented here for comparison. Bands were spliced from the same gel for representation purposes. (**D**) Model for how the two ATPase active sites regulate cohesin function. Free cohesin is converted to a stable DNA-bound form by the action of the loader complex (not shown) and ATP binding/hydrolysis by ATPase active sites. A particular unknown nucleotide state at the Smc3 ATPase active site induces the tethering (cohesive) form. Upon finding the correct sister to be paired with, the Eco1-mediated acetylation of Smc3 leads to the stabilization of this cohesive state. Absent this stabilization, the Smc3 ATPase active site destabilizes the tethering form or induces cohesin dissociation from chromosomes. Wpl1 could either promote this Smc3 ATPase active site function or destabilize the cohesin bound to DNA in the non-tethering form. Figure supplement 1 shows cohesin binding to DNA in *smc1-D1164E* and *smc3-D1161E* strains.

The following figure supplement is available for figure 7:

Figure 7 continued

**Figure supplement 1.** Chromosomal binding of *smc1-D1164A* and *smc3-D1161A* mutants assayed with antibodies against tagged Smc proteins.

## Discussion

### The cohesin ATPase functions in tethering DNA after cohesin's stable binding to DNA

Until this study, there was no evidence for a role for the ATPase regulating post-DNA binding steps. Here, we show that cohesin retains its ATPase activity after stably binding to DNA in vitro, implying ATP hydrolysis plays an additional role(s) after cohesin binds DNA. We also show that cohesion activator mutations in key residues of the Smc3 ATPase active site, *smc1-D1164E* and *smc1-Y1128C*, suppress the severe cohesion defect of *eco1Δ wpl1Δ* cells, similar to the previously described *smc3-D1189H* allele (*Guacci et al., 2015*). These cohesion activators restore cohesion in *eco1Δ wpl1Δ* cells without increasing the amount of cohesin stably bound to DNA (this study, *Guacci et al., 2015*). Lastly, a mutation analogous to one of these activator mutations, *smc1-D1164E*, abolishes most, if not all, cohesin ATPase activity when introduced in purified *S. pombe* cohesin complex. Taken together, our results suggest the robust ATPase activity of DNA-bound cohesin is physiologically relevant, likely acting as an inhibitor of the conversion of stably bound cohesin to a form capable of tethering sister chromatids.

### Smc3 ATPase active site has a unique function in DNA tethering

We also show that the cohesion activator phenotype of the *smc1-D1164E* allele is a unique feature of the D-loop of the Smc3 ATPase active site. The analogous substitution in the D-loop of the Smc1 ATPase active site, *smc3-D1161E*, not only fails to act as a cohesion activator but also is unable to support viability, fails to load cohesin to DNA, and cannot generate cohesion. These results suggest that the Smc1 and Smc3 ATPase active sites differentially affect cohesin ATPase function and that the Smc3 ATPase active site has a distinct function in the generation of cohesion after stable DNA binding of cohesin. This distinct function provides an explanation for the evolutionary diversion of the SMC subunits. Indeed, use of asymmetrical ATPases for regulation purposes is common to ABC transporters (*Antony and Hingorani, 2004*; *Furman et al., 2013*; *Yang et al., 2003*). We suspect the differential roles of cohesin ATPase active sites and a specialized role for the Smc3 active site in DNA tethering was missed previously, in part due to their composite nature and the particular mutations that have been analyzed to date. How the D-loop of the Smc3 ATPase active site executes its specialized function in regulating the cohesin ATPase remains unclear. The D-loop may impact the nucleotide state only in the Smc3 ATPase active site. Alternatively, it could alter the nucleotide state of both active sites. Support for the latter comes from studies on the ABC ATPase domains in ABC transporters, which suggest that ATP hydrolysis by the two sites is cooperative (*Holland and Blight, 1999*), and that D-loops are involved in mediating the communicat on between the two active sites within the ATPase (*Furman et al., 2013*; *Hohl et al., 2012, 2014*).

### Cohesin functions can be separated from overall ATPase activity levels

The phenotypes of the D-loop mutants in the Smc1 and Smc3 active sites did not correlate with the ATPase activities of the corresponding purified *S. pombe* cohesin complexes. For example, cohesin complexes with aspartate-to-glutamate substitution in the D-loop of the Smc1 active site had significant ATPase activity in vitro, but were severely perturbed for DNA binding in *S. cerevisiae*. In addition, cohesin complexes with aspartate-to-glutamate substitution in the D-loop of the Smc3 ATPase had low levels of ATPase activity indistinguishable from the walker A mutant, but could promote DNA binding and cohesion in *S. cerevisiae*. While it is possible that this lack of correlation could be explained by our use of the ATPase activity of the *S. pombe* wild type and mutant cohesins as an approximation of the *S. cerevisiae* cohesin, this explanaition seems unlikely. The aspartate of the D-loop is absolutely conserved in all SMC complexes and highly conserved among very diverse ABC ATPase modules. Our analysis of analogous substitutions in the homodimeric Rad50 protein shows similar reductions in the ATPase activity. Finally, many published biochemical and structural studies have used cohesin complexes from different organisms as a proxy for *S. cerevisiae* cohesin to

provide physiologically relevant insights. Intriguingly, the uncoupling of ATPase activity and function by D-loop mutations has been observed for several other ABC ATPases (*Furman et al., 2013*; *Grossmann et al., 2014*).

Therefore, we favor a model in which cohesin's functions and conformations could be coordinated by the different states in the hydrolysis cycle of ATPase active sites. For example, the tethering-competent state of cohesin could require that the Smc3 active site be bound with ADP+Pi. The *smc1-D1164E* mutation may mimic the ADP-Pi state, whereas the *smc1-D1164A* mutant could be blocked elsewhere in the ATPase cycle (ATP bound or nucleotide free). Both the *smc1-D1164E* and *smc1-D1164A* would dramatically reduce overall ATPase activity but only the *smc1-D1164E* would promote cohesin function. A similar case of trapping the ADP+Pi state is hypothesized for the MRP1 ABC transporter, for a glutamate substitution of an aspartate residue (*Qin and Cao, 2008*). Elucidating the exact relationship between the particular nucleotide states of the Smc1 and Smc3 ATPase active sites and cohesin functions will likely require in vitro reconstitution of cohesin's various functions and detailed structural studies.

## Eco1 and Wpl1 regulate cohesin's tethering activity through cohesin ATPase

The phenotypes of *smc1-D1164E* cells suggest that Eco1 modulation of cohesin's ATPase impacts two distinct modes of cohesin regulation. First, *smc1-D1164E* suppresses the inviability of *eco1Δ* cells, which normally die because they are unable to inhibit Wpl1 (*Rolef Ben-Shahar et al., 2008*; *Rowland et al., 2009*; *Sutani et al., 2009*). This result suggests that Eco1 acetylation alters the Smc3 ATPase active site to make cohesin resistant to Wpl1. Second, *smc1-D1164E* also suppresses the cohesion defect in *eco1△ wpl1△* cells, suggesting that it phenocopies the second function of Eco1 acetylation, the activation of DNA-bound cohesin to tether sister chromatids.

A clue to how Eco1 modulates cohesin ATPase to promote DNA tethering comes from our in vitro and in vivo characterization of *smc1-D1164E*. Since *smc1-D1164E* suppresses the *eco1Δ* cohesion defect, we propose that *smc1-D1164E* mimics the Eco1 acetylation of Smc3 by blocking the cohesin ATPase at a particular stage in its hydrolysis cycle, promoting DNA tethering. In the absence of acetylation, continued cycles of ATP hydrolysis prevent cohesin from maintaining the conformation necessary for tethering (Figure 7D). The completion of the ATPase cycle may render cohesin in a conformation capable of being recognized by and acted upon Wpl1, which leads to cohesin dissociation from DNA.

This model predicts that Eco1-mediated acetylation of the Smc3 may have an impact on the ATPase activity of cohesin. However, previous studies on in vitro acetylated cohesin or acetyl-mimic mutants of cohesin suggest that Eco1 acetylation does not impact cohesin's total ATPase activity (*Ladurner et al., 2014*). This could be due to (1) the inherent inefficiency of acetylation both in vitro and in vivo, resulting in only a small subset of cohesin being acetylated, (2) the inability of glutamine substitutions of K112 and K113 in Smc3 to mimic the acetylated state, (3) the failure to assay cohesin ATPase activity on DNA. Alternatively, Eco1-mediated acetylation may stabilize an intermediate state of cohesin in a way that makes subsequent cycles of ATP hydrolysis unable to alter the tethering conformation. Either way, the result is that acetylation traps cohesin in its tethering form to stabilize cohesion during or soon after establishment. Testing whether Eco1-mediated acetylation directly downregulates the cohesin ATPase cycle when bound to DNA and how the ATPase cycle affects cohesin conformation will require better reconstitution of cohesin function in vitro.

Our model of transient conformational changes that must be stabilized provides a plasticity to cohesin-mediated DNA tethering. Cohesin that is bound to a locus could form transient tethers with a number of different DNA loci and at different times. The spatial and temporal control of Eco1 would ensure that only the proper tethers are stabilized, allowing a very tight regulation for cohesion establishment in sister chromatid cohesion. This plasticity of cohesin tethering and its Eco1-dependent regulation may be critical for other aspects of cohesin function, such as condensation, DNA repair and transcriptional regulation.

## Materials and methods

### Protein expression and purification

*S.pombe* cohesin complex was over-expressed and purified from *S. cerevisiae* as described (*Murayama and Uhlmann, 2014*) with minor modifications. Briefly, over-expressing cells were harvested and lysed with a cryomill. For cohesin purifications, 50 mL of cell powder was thawed in the presence of 100 mL CLH300 buffer (50 mM HEPES, pH 7.5, 1 mM DTT, 300 mM NaCl, 20% glycerol, and protease inhibitor cocktail from Roche) and the clarified by centrifugation. The clarified supernatant was bound to 2 mL IgG beads (Invitrogen) overnight in the presence of RNase A (10 µg/mL). The beads were washed 300 mL in H300 (300 mM NaCl, 25 mM HEPES pH 7.5, 0.5 mM TCEP, 10% glycerol). To elute protein off the beads, 5 mL of elution buffer (H300 with 2 mM $MgCl_2$, 5 U/mL Prescission protease, 10 µg/mL RNase A) was added. Eluted protein was diluted to achieve 100 mM NaCl and then bound to heparin column. The column was washed in 4xColumn Volume (CV) H100M (25 mM HEPES pH 7.5, 0.5 mM TCEP, 10% glycerol, 100 mM NaCl and 2 mM $MgCl_2$), 4xCV H300 with 2 mM $MgCl_2$. Cohesin was eluted from the column at 600 mM NaCl. Cohesin containing fractions were pooled, concentrated and frozen in same buffer containing 200 mM NaCl. Mutants of cohesin were generated by site-directed mutagenesis and over-expression strains were generated as described (*Murayama and Uhlmann, 2014*). Mutant proteins were purified using the same protocol as described above. The loader complex from *S.pombe* was expressed and purified as described (*Murayama and Uhlmann, 2014*). Rad50 protein from T4 bacteriophage was purified as described (*la Rosa and Nelson, 2011*).

### Assembly of stable cohesin-DNA complexes in vitro

*CARC1* DNA substrates were prepared as described (*Onn and Koshland, 2011*). Briefly, for each binding reaction, 500 ng biotin-labeled DNA was assembled on 20 µL streptavidin-conjugated dynabeads. Beads were washed 6x 30 µL in CL1 buffer (35 mM Tris pH 7.5, 1 mM TCEP, 25 mM NaCl, 25 mM KCl, 1 mM $MgCl_2$, 15% glycerol and 0.003% Tween 20) and resuspended in 60 µL CL1. 80 nM cohesin complex and 0.5 mM ATP (with or without 80 nM loader) was added to beads and incubated at 30°C for 1 hour in a reaction volume of 100 µL. Assembled cohesin-DNA complexes were washed in 100 µL buffer once in CL1, three times in CL1 with 500 mM KCl, and once more with CL1. Resuspending beads in 30 µL SDS sample buffer eluted bead-bound cohesin-DNA complexes. Samples were analyzed by SDS-PAGE followed by Coomassie staining or western blotting.

### Elution of cohesin from DNA-beads with DNase or restriction enzyme digests

Stable cohesin-DNA complexes assembled as described above were resuspended in CL1 buffer in the presence of 2U DNase or 5U Mnl I at 30°C for 30 minutes. Eluted proteins (supernatant) and beads were subjected to SDS-PAGE and proteins were visualized by Coomassie staining or Western blotting.

### ATPase assays

For basal ATPase activity of cohesin, 150 nM cohesin was diluted in 100 µL ATPase reaction buffer 1 (25 mM HEPES pH 7.5, 100 mM NaCl, 10% glycerol, 1 mM TCEP, 5 mM $MgCl_2$). 10 µL of this sample was run on SDS-PAGE to visualize cohesin used in ATPase experiments. The ATPase reaction was started by the addition of 20 µL of 5x ATP hot mix (2.5 mM ATP, 1 µL of 10 µCi/ µL ATP-γ-P32 in 96.5 µL 5xATP buffer) to 80 µL cohesin mix. Thus, in our ATPase reactions, the final concentration of cohesin was 120 nM, and ATP was 0.5 mM. At appropriate time points, 20 µL of this reaction mix was taken out and mixed with 380 µL of 0.25 mg/mL BSA. 300 µL of STOP buffer (100 µL $KPO_4$ 1 M, 100 µL 1N HCl, 100 µL 20% Norit) was immediately added to the samples. Samples were then spun at 4°C 10000 rpm for 3 minutes. 500 µL of the supernatant was taken out in a new tube and spun again at 4°C 10000 rpm for 3 minutes. 350 µL of the supernatant of this second spin was counted in 5 mL of scintillation cocktail using a scintillation counter. Reads at time zero were counted as background and subtracted from later time points. To get% ATP hydrolysis, 10 µL of 1:5 diluted 5xATP buffer was counted to represent 100% ATP hydrolysis. For loader- and DNA-stimulated ATPase activity of cohesin, 80 nM cohesin, 80 nM loader, and 2.5 µg plasmid DNA containing the sequence

for *CARC1* (pIO2) were incubated in 50 μL ATPase reaction buffer 2 (25 mM HEPES pH 7.5, 25 mM NaCl, 25 mM KCl, 10% glycerol, 1 mM TCEP, 5 mM MgCl$_2$). To measure the activity of stably DNA-bound cohesin complexes (CD-B), 80 nM cohesin, 80 nM loader and 0.5 mM ATP was incubated with 100 μL dynabeads coupled to 2.5 μg DNA in 500 μL total volume of CL1 at 30°C for 1 hour. Unbound/unstably bound cohesin was washed off using the following washes: 1x500 μL CL1, 3 x 500 μL in CL1 with 500 mM KCl, then 1x500 μL CL1. CD-B were then resuspended in 50 μL ATPase buffer 2. This sample of CD-B contained about% 20 of the initial input cohesin, which was approximately the same amount of cohesin as the other samples in this experiment (*Figure 1E*), as could be judged by the protein gel. The reaction was started by the addition of 10 μL 5xATP hot mix to 40 μL protein/DNA mix for 2 hours at 30°C and samples were processed as described above. The ATPase activity of T4 Rad50 and mutants was measured in ATPase buffer 1 for 2 hours at 30°C as described above, except 1 μM Rad50 was used. Error bars represent standard error of data from at least two independent experiments.

## Yeast strains and media

Yeast strains used in this study are A364A background, and their genotypes are listed in *Supplementary file 1*. SC minimal and YPD media were prepared as described (*Guacci et al., 1997*). Benomyl (a gift from Dupont) and camptothecin (Sigma) plates used to assess drug sensitivity were prepared as previously described (*Guacci and Koshland, 2012*). Preparation of auxin (Sigma) containing media for depletion of AID tagged proteins was as previously described (*Eng et al., 2014*).

## Dilution plating assays

Cells were grown to saturation in YPD media at 23°C then plated in 10-fold serial dilutions. Cells were incubated on plates at relevant temperatures or containing drugs as described. For plasmid shuffle assays, cells were grown to saturation in YPD media to allow loss of covering plasmid, then plated in 10-fold serial dilutions on YPD or FOA media.

## G1 arrest and release into M phase arrest

### G1 arrest

Asynchronous cultures of cells were grown to mid-log phase at 23°C in YPD media, then α factor (Sigma) added to 10$^{-8}$ M. Cells were incubated for 3 h to induce arrest in G1 phase. This incubation time was increased to 3.5 h for all strains in any experiment where an *eco1Δ wpl1Δ* background strain was used. For depletion of AID tagged proteins, auxin was added (500 mM final) to G1 arrested cells, and then incubated an additional 1 h in α factor containing media.

### Release from G1 into M phase arrest

G1 arrested cells were washed 3x in YPD containing 0.1 mg/ml Pronase E (Sigma), once in YPD, then resuspended in YPD containing nocodozale (Sigma) at 15 μg/ml final. Cells were incubated at 23°C for 3 h to arrest in M phase. For depletion of AID tagged proteins, auxin was added (500 mM final) in all wash media and in resuspension media containing nocodozole to ensure depletion at all times.

## Monitoring cohesion using LacO-GFP assay

Cohesion was monitored using the LacO-LacI system where cells contained a GFP-LacI fusion and tandem LacO repeats integrated at one chromosomal locus, which recruits the GFP-LacI (*Straight et al., 1996*). *CEN*-distal cohesion was monitored by integrating LacO repeats at *LYS4*, located 470 kb from *CEN4*. *CEN*-proximal cohesion is monitored by integrating LacO at *TRP1*, located 10 kb from *CEN4*. Cells were fixed, and processed to allow the number of GFP signals in each cell to be scored and the percentage of cells with 2-GFP spots determined as previously described (*Guacci and Koshland, 2012*). Data is from 2 independent experiments and 200-300 cells scored for each data point in each experiment.

## Plasmid constructs

Site directed mutagenesis using the Stratagene Quick-change kit was employed to generate all mutants described. Mutations were confirmed by sequencing the entire ORF to ensure it was the only change.

## Strain construction

### SMC1 and SMC3 Shuffle strain construction

Haploids containing pTH2 (*SMC1 URA3 CEN*) or pEU42 (*SMC3 URA3 CEN*) plasmid had their endogenous *SMC1* or *SMC3* gene deleted and replaced by the HPH cassette (encodes resistance to Hygromycin B) using standard PCR mediated homology-based recombination.

### Plasmid shuffle to assess *SMC1* and *SMC3* mutant "test" alleles by integration at the *LEU2* locus

*SMC1 or SMC3* WT of mutant alleles were cloned onto an integrating *LEU2* vector. For *SMC1* test alleles, these were plasmid pVG444 (*SMC1 LEU2*), pVG456 (*smc1-D1164E LEU2*) and pVG4 (*smc1-D1164A LEU2*), linearized within *LEU2* by PpuMI. For *SMC3* test alleles, these were plasmid pVG419 (*SMC3 LEU2*), pVG419 D1161A (*smc3-D1161A LEU2*) and pVG458 (*smc3-D1161E LEU2*) linearized within *LEU2* using BstEII. Linearized plasmids were transformed into an *SMC1* or *SMC3* shuffle strain to integrate the vector at the *LEU2* locus and LEU+ clones isolated. These "test alleles" were assayed for their ability to support viability as the sole *SMC1* or *SMC3* as sole source as follows. LEU + clones were grown to saturation in YPD media at 23°C to allow loss of either plasmid pTH2 or pEU42, then plated in 10-fold serial dilutions on media containing 5-fluoroorotic acid (FOA). FOA selectively kills *URA3* cells, thereby selecting for loss of pTH2 or pEU42, which allows assessment of test allele ability to support viability as the sole *SMC1 or SMC3* in cells, respectively. As a control, cells were also plated on either YPD or URA- media.

### Insertion of *SMC1* and *SMC3* alleles at the endogenous locus

Two different strategies were employed. One utilized shuffle strains described above for one step-gene replacement. A linear DNA fragment containing the desired *SMC1* or *SMC3* ORF, promoter and 3' UTR were transformed into shuffle strains, plated on YPD and grown overnight. Plates were replica plated to FOA and FOA resistant clones selected, tested for sensitivity to Hygromycin B, which occurs when *smc1Δ::HPH* or *smc3Δ::HPH* is replaced by the transformed linear *SMC1* or *SMC3* allele, respectively. Transplacement alleles were confirmed by PCR screening and PCR sequencing.

The second strategy used to insert *SMC1* and *SMC3* alleles at the endogenous locus in haploid *eco1Δ wpl1Δ* cells, *wpl1Δ* cells, or in WT cells bearing *TIR1* was as follows. Plasmids encoding the desired alleles were linearized within the ORF. Linearized plasmids were transformed into haploid strains. URA+ colonies contain the *SMC1 URA3* or *SMC3 URA3* plasmid integrated at the *SMC1* or *SMC3* locus to create tandem *SMC1* or *SMC3* genes. URA+ transformants were replica plated onto YPD then dilution streaked on FOA to excise the *URA3* marker, and thereby select for loss of one *SMC1/3* allele. PCR mediated sequencing was used to identify clones containing only the desired alleles.

## Strains containing *AID* tagged proteins

Details about the Auxin mediated destruction of AID tagged proteins in yeast was previously described (*Eng et al., 2014*). Briefly, the *TIR1* E3-ubiquiting ligase placed under control of the GPD promoter and marked by *C. Albicans TRP1* replaced the *TRP1* gene on chromosome IV. *SMC1*, *SMC3* and *ECO1* were internally tagged with *3V5-AID2* sequences and transformed into yeast strains bearing *TIR1* to replace *SMC1*, *SMC3* and *ECO1* at the endogenous locus. PCR screening and auxin mediated sensitivity were used to identify clones containing *AID* tagged genes.

**Genetic screen of *eco1Δ wpl1Δ* cells for cohesion activator suppressors** was done as previously described (*Guacci et al., 2015*).

**Chromatin Immunoprecipitation** (**ChIP**) was performed as previously described (*Eng et al., 2014*; *Wahba et al., 2013*).

**Microscopy**. Images were acquired with a Zeiss Axioplan2 microscope (100X objective, NA=1.40) equipped with a Quantix CCD camera (Photometrics).

**Flow cytometry analysis** was performed as previously described (*Eng et al., 2014*).

## Acknowledgements

We thank Hugo Tapia, Thomas Eng, Jeremy Amon, Michelle Bloom, Orna Cohen-Fix and Jill Heidinger-Pauli for critically reading the manuscript, and other members of the Koshland lab for fruitful discussions. We thank Dr. Frank Uhlmann for sharing cohesin and loader expression strains and plasmids, Dr. Scott Nelson for sharing Rad50 expression plasmids, Dr. Stuart Lynn for advice on ATPase assays, the laboratory of David Drubin and Georjana Barnes for use of equipment. This work was funded by the National Institute of Health (GM092813). GC was supported by a fellowship from the Damon Runyon Cancer Research Foundation (DRG-2137-12).

## Additional information

### Funding

| Funder | Grant reference number | Author |
|---|---|---|
| National Institutes of Health | GM092813 | Douglas Koshland |
| Damon Runyon Cancer Research Foundation | DRG-2137-12 | Gamze Çamdere |

The funders had no role in study design, data collection and interpretation, or the decision to submit the work for publication.

### Author contributions

GÇ, VG, Conception and design, Acquisition of data, Analysis and interpretation of data, Drafting or revising the article, Contributed unpublished essential data or reagents; JS, Acquisition of data, Analysis and interpretation of data, Drafting or revising the article; DK, Conception and design, Drafting or revising the article

## Additional files

### Supplementary files

• Supplementary file 1. Strain table.

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
