## [Decision Letter]

Thank you for submitting your work entitled "The ATPases of cohesin interface with regulators to modulate cohesin-mediated DNA tethering" for consideration by *eLife*. Your article has been reviewed by two peer reviewers, and the evaluation has been overseen by a Reviewing Editor and James Kadonaga as the Senior Editor.

The reviewers have discussed the reviews with one another and the Senor Editor has drafted this decision to help you prepare a revised submission.

Cohesins belong to the family of SMC containing protein complexes that play a critical role in chromosome structure and sister chromatid cohesion, the process that holds the replicated DNA strands together until their segregation in anaphase. This manuscript addresses how the SMC ATPases function in this process.

Previous studies suggested that the two ATPases of cohesin, Smc1 and Smc3, are critical for cohesins to bind DNA. This study addresses this possibility in mechanistic detail and suggests that only one of the ATPases, Smc1, is required for this process and that the other ATPase, Smc3 is required for cohesion establishment after cohesin binds to the DNA.

The findings presented are interesting, but important aspects of the conclusions need to be validated. We would therefore be interested in a revised manuscript that suitably addresses the points outlined below.

1) The authors should cite appropriate references for the schematic diagram for the Smc1-Smc3 head domain interaction in Figure 1 and stipulate that this has not been experimentally proven for cohesin Smc1 and Smc3.

2) It is interesting that DE mutations in Smc1 and Smc3 at equivalent amino acid residues have distinct effects on ATPase activity, cell viability, and cohesin binding to chromatin. However, no experimental data were provided to demonstrate, as summarized in Figure 7, that Smc1DE and Smc3DE mutations only affect Smc3 and Smc1 ATPases, respectively. Therefore, the authors should test whether Smc1DE affects Smc3 ATPase, and Smc3DE affects Smc1 ATPase. If this is not possible, the conclusions and model figure should be appropriately adjusted to match the supporting data.

3) Apparent discordance between the mutants' ability to support cohesion/viability and their ATPase activities is confusing. Additional data should be provided by performing both in vivo and in vitro studies using the same species. Alternatively, the authors should acknowledge the possible problem of comparing *S. pombe* cohesin for the in vitro studies and *S. cerevisiae* cohesin for the in vivo studies and tone down the conclusion.

4) Figure 4—figure supplement 3, Figure 5, and Figure 5—figure supplement 1, Figure 6 and Figure 7: The labeling "Smc3 ATPase" and "Smc1 ATPase" for Smc1-D1164 and Smc3-D1161 mutants are confusing and potentially misleading without further proof and thus, should be deleted unless additional data are provided to validate these conclusions (as in point #2).

---

## [Author Response]

*1) The authors should cite appropriate references for the schematic diagram for the Smc1-Smc3 head domain interaction in Figure 1 and stipulate that this has not been experimentally proven for cohesin Smc1 and Smc3.*

We have included the appropriate references in the figure legend and main text, and added that this is a model based on the available Rad50 and Smc1 head domain structures in presence of ATP, and that this model is not experimentally proven for the heterodimeric cohesin ATPase head domain. We also have added the following definition sentence to our Introduction to clarify the composite nature of the cohesin ATPase head domain: “Cohesin’s ATPase domain is composed of two ATPase active sites each containing four conserved motifs: Walker A, Walker B, signature motif, and D-loop. In the Smc1 ATPase active site the Walker A and Walker B motifs are encoded by Smc1, while the D-loop and signature motifs are encoded by Smc3. Likewise, the Smc3 ATPase active site contains the Smc3-encoded Walker A and Walker B motifs, and Smc1-encoded D-loop and signature motifs (Arumugam et al., 2006; Haering et al., 2004; Hopfner et al., 2000).”

*2) It is interesting that DE mutations in Smc1 and Smc3 at equivalent amino acid residues have distinct effects on ATPase activity, cell viability, and cohesin binding to chromatin. However, no experimental data were provided to demonstrate, as summarized in Figure 7, that Smc1DE and Smc3DE mutations only affect Smc3 and Smc1 ATPases, respectively. Therefore, the authors should test whether Smc1DE affects Smc3 ATPase, and Smc3DE affects Smc1 ATPase. If this is not possible, the conclusions and model figure should be appropriately adjusted to match the supporting data.*

We thank the reviewers and agree that although the D-loop residues in question form part of pme active site, there is no evidence that our D loop mutants only affect one ATPase active site. Given the proposed role of the D-loops in mediating communication between the two active sites, it is certainly possible that they would have an effect on the ATPase activity of both subunits. Indeed, as shown before, the walker A mutations in either subunit appears to effect activity of both subunits, bringing down the total activity to near background levels, further arguing that it is very difficult to affect one ATPase without affecting the other in ABC ATPases. As mentioned by the reviewers, our main point is that the two ATPase active sites are not functionally equivalent, suggesting that they modulate different steps in cohesin function. To clarify this point, we have now altered our model in Figure 7, and expanded our Discussion (please see below) to mention that we refer to the regulation of the cohesin ATPase by these active sites, and that we are not measuring the individual activities of each active site.

“How the D-loop of the Smc3 ATPase active site executes its specialized function in regulating the cohesin ATPase remains unclear. The D-loop may impact the nucleotide state only in the Smc3 ATPase active site. Alternatively, it could alter the nucleotide state of both active sites. Support for the latter comes from studies on the ABC ATPase domains in ABC transporters, which suggest that ATP hydrolysis by the two sites is cooperative (Holland and Blight, 1999), and that D-loops are involved in mediating the communication between the two active sites within the ATPase (Furman et al., 2013; Hohl et al., 2012; 2014).”

*3) Apparent discordance between the mutants' ability to support cohesion/viability and their ATPase activities is confusing. Additional data should be provided by performing both in vivo and in vitro studies using the same species. Alternatively, the authors should acknowledge the possible problem of comparing* S. pombe *cohesin for the in vitro studies and* S. cerevisiae *cohesin for the in vivo studies and tone down the conclusion.*

We address the reviewers’ concerns in several ways. First, we now acknowledge that using ATPase activity of the *S. pombe* wild-type and mutant cohesins is an approximation of the *S. cerevisiae* cohesin and some of the discordance could possibly be explained by this approximation. We present our arguments for why the use of the *S. pombe* cohesin as a proxy and the observed discordance is likely physiologically relevant. Finally, we improve the presentation of our model that the intermediates in the ATPase cycle are critical for regulating cohesin function, not the level of ATPase activity. A mutant protein that blocks the exchange of ADP-Pi with ATP will dramatically reduce the level of ATPase activity, but it would not impair cohesin function if the ADP-Pi state is the active form. In contrast, a mutant that radically reduces ATP hydrolysis will also reduce ATPase activity but trap cohesin in the ATP-bound state, which could impair cohesin function if this state is inactive. We hope that these revisions will make the discordance enlightening and no longer confusing. We have incorporated these points into the Discussion under the subheading “Cohesin functions can be separated from overall ATPase activity levels”.

*4) Figure 4—figure supplement 3, Figure 5, and Figure 5—figure supplement 1, Figure 6 and Figure 7: The labeling "Smc3 ATPase" and "Smc1 ATPase" for Smc1-D1164 and Smc3-D1161 mutants are confusing and potentially misleading without further proof and thus, should be deleted unless additional data are provided to validate these conclusions (as in point #2).*

We thank the reviewers for pointing out this potentially misleading wording. We included a section in the Introduction to clarify the definition of the two ATPase active sites in cohesin’s ATPase head domain (please see our response to point 1). We have also altered the wording in the mentioned figures and in the manuscript, referring to the sites as Smc1 and Smc3 ATPase active sites (a published convention as used in Arumugam et al. 2006). This change emphasizes location of the mutations in the D-loops without inferring a conclusion about the impact of the mutation on the specific ATPases.